# Advancing Analytic Class-Incremental Learning through Vision-Language Calibration

**Binyu Zhao** [1]   **Wei Zhang** [1]   **Xingrui Yu** [2 3]   **Zhaonian Zou** [1]   **Ivor Tsang** [2 3 4]

## Abstract

Class-incremental learning (CIL) with pre-trained models (PTMs) faces a critical trade-off between efficient adaptation and long-term stability. While analytic learning enables rapid, recursive closed-form updates, its efficacy is often compromised by accumulated errors and feature incompatibility. In this paper, we first conduct a systematic study to dissect the failure modes of PTM-based analytic CIL, identifying representation rigidity as the primary bottleneck. Motivated by this insight, we propose **VILA**, a novel dual-branch framework that advances analytic CIL via a two-level vision-language calibration strategy. Specifically, we coherently fuse plastic, task-adapted features with a frozen, universal visual anchor at the feature level through geometric calibration, and leverage cross-modal semantic priors at the decision level to rectify prediction bias. This confluence maintains analytic-learning's extreme efficiency while overcoming its inherent brittleness. Extensive experiments across eight benchmarks demonstrate that VILA consistently yields superior performance, particularly in fine-grained and long-sequence scenarios. Our framework harmonizes high-fidelity prediction with the simplicity of analytic learning. Our code is available at https://github.com/byzhaoAI/VILA.

## 1. Introduction

The capacity to acquire knowledge consecutively from non-stationary data streams is foundational for intelligent sys-

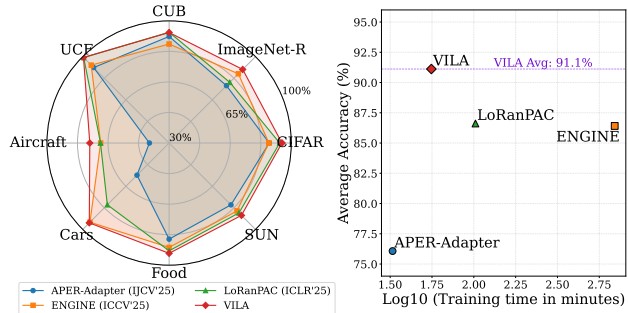

*Figure 1.* Performance and efficiency overview. (Left) Performance comparison across 8 diverse benchmarks. It demonstrates VILA's superior generality in both coarse- and fine-grained scenarios. (Right) Training time *vs.* Average accuracy. VILA occupies the optimal top-left corner, offering high-fidelity predictions with significantly lower latency.

tems operating in dynamic real-world environments. Ideally, such systems must efficiently integrate new concepts while preserving previously mastered skills (McCloskey & Cohen, 1989). However, standard neural networks suffer from catastrophic forgetting when trained on shifting data distributions. While class-incremental learning (CIL) (Masana et al., 2022) aims to address this capability, achieving an optimal balance between stability (retaining old knowledge) and plasticity (learning new knowledge) remains a challenging objective.

Recent advancements have leveraged pre-trained models (PTMs) as a cornerstone strategy. PTMs provide robust, transferable feature representations that significantly elevate the performance baseline of CIL. However, the majority of PTM-based methods typically rely on iterative optimization of learnable parameters (*e.g.*, via adapters or prompt tuning) (Wang et al., 2022b;a; Liang & Li, 2024; Roy et al., 2024; Yu et al., 2025; Wang et al., 2025a). Although these methods have improved CIL performance, they are often prone to performance degradation as the number of tasks accumulates, driven by the progressive perturbation of the representation space by gradient updates. Moreover, some of them can be cumbersome to execute, requiring intricate training procedures and meticulous hyperparameter tuning to seek a delicate stability-plasticity equilibrium, which hinders their deployment in efficiency-critical scenarios.

---

[1]School of Computer Science and Technology, Harbin Institute of Technology, China [2]CFAR, Agency for Science, Technology and Research, Singapore [3]IHPC, Agency for Science, Technology and Research, Singapore [4]College of Computing and Data Science, Nanyang Technological University, Singapore. Correspondence to: Wei Zhang <weizhang@hit.edu.cn>, Xingrui Yu <yu_xingrui@a-star.edu.sg>.

*Proceedings of the 43$^{rd}$ International Conference on Machine Learning*, Seoul, South Korea. PMLR 306, 2026. Copyright 2026 by the author(s).

Analytic learning (Guo et al., 2001; Zhuang et al., 2021) emerges as a compelling alternative path for CIL. By utilizing closed-form recursive least squares (RLS) updates, analytic learning transforms the learning process into a non-iterative matrix operation, offering mathematically optimal weights with theoretically zero forgetting of the learned mapping. Consequently, combining analytic learning with PTMs presents a promising direction (McDonnell et al., 2023; Peng et al., 2025; Gao et al., 2025). It aims to maintain high-level CIL performance by leveraging strong pre-trained features; Meanwhile, it alleviates the execution complexity and computational costs inherent to iterative methods.

Despite these advantages, we conduct a systematic analysis of this paradigm and identify a critical bottleneck termed *representation rigidity* that restricts its full potential. Our analysis suggests that while analytic learning effectively preserves historical knowledge within a given subspace, the feature space itself (typically frozen or narrowly adapted) may become rank-deficient for future distinct distributions. This implies that although analytic learning mitigates forgetting at the classifier level, it may lack the necessary feature plasticity to fully accommodate novel semantic clusters, potentially leading to accumulated approximation errors in long-term learning.

Based on this insight, we design **VI**sion-**L**anguage **A**nalytic learning (**VILA**), an asymmetric dual-branch framework to address the rigidity issue. VILA employs a two-level calibration strategy to integrate the discriminative plasticity of a task-adapted Vision Transformer (ViT) with the generalizable stability of a frozen vision-language model (VLM). At the feature level, it geometrically aligns heterogeneous features onto a unified manifold. At the decision level, it utilizes semantic priors to rectify decision boundaries. Through this dual calibration, VILA effectively constructs a robust semantic subspace compatible with recursive updates. As shown in Figure 1, VILA demonstrates leading performance across 8 benchmarks, consistently outperforming iterative baselines with significantly higher training efficiency. The main contributions are summarized as follows:

- We provide a systematic analysis of PTM-based analytic CIL. Our investigation exposes representation rigidity as a key bottleneck, and highlights the generalization constraints of single-branch architectures.

- We propose VILA, which integrates analytic learning with a frozen VLM via geometric and semantic calibration, effectively spanning a robust subspace to support continuous analytic updates.

- Extensive experiments demonstrate that VILA achieves superior performance and efficiency, offering a practical and robust solution for online class-incremental learning.

## 2. Related Works

**Class-incremental learning (CIL).** CIL is designed to continuously accumulate new knowledge from sequential data streams without erasing previously acquired capabilities. Traditional approaches generally fall into three paradigms. *Regularization-based* methods (Kirkpatrick et al., 2017; Li & Hoiem, 2017) impose constraints on weight updates to preserve important parameters of prior tasks. *Replay-based* strategies (Rebuffi et al., 2017; Buzzega et al., 2020) maintain a memory buffer of historical exemplars to rehearse previous knowledge. *Architecture-based* approaches (Fernando et al., 2017; Mallya & Lazebnik, 2018) dynamically expand the network capacity to isolate task-specific parameters. While foundational, these methods often struggle to balance scalability and performance in large-scale settings.

**PTM-based CIL.** Traditional CIL methodologies were predominantly centered on training models from scratch. In contrast, recent advances have shifted this paradigm towards adapting PTMs. The dominant strategy is parameter-efficient fine-tuning (PEFT), which includes three parallel streams: *Prompt-based* methods (Wang et al., 2022b;a) that insert learnable tokens to instruct attention; *Adapter-based* methods (Wang et al., 2025b; Zhou et al., 2025a; Gao et al., 2025) that inject bottleneck layers for feature modulation; and *LoRA-based* strategies (Hu et al., 2022; Liang & Li, 2024; Wu et al., 2025; He et al., 2025) that optimize low-rank updates. Despite their effectiveness, these iterative methods incur significant latency due to backpropagation.

Furthermore, VLMs have introduced a new dimension by leveraging large-scale image-text pre-training to align visual features with rich linguistic semantics. This paradigm shifts the focus from closed-set classification to open-world recognition. While VLMs like CLIP (Radford et al., 2021) possess strong zero-shot classification capabilities, they often lack domain specialization and struggle with fine-grained distinctions (*e.g.*, specific aircraft variants or medical conditions). To address this, recent VLM-based works (Yu et al., 2024; Ma'sum et al., 2025; Yu et al., 2025; Zhou et al., 2025b) aim to adapt these models to continuous streams. However, they typically rely on heavy tuning, or fail to fully exploit the geometric complementarity between visual adaptation and frozen semantic priors.

**Analytic CIL.** Distinct from gradient-based optimization, analytic learning converts training into a regularized least-squares problem, yielding closed-form solutions via recursive updates. Recent advances (McDonnell et al., 2023; Peng et al., 2025) successfully scale this paradigm to PTMs by projecting high-quality deep features into a high-dimensional buffer. This approach offers theoretically guaranteed stability and extreme efficiency. However, the limitation of representation rigidity still remains. Since the analytic solver requires a fixed input distribution, existing

*Table 1.* Pilot study on *representation rigidity*. Performance comparison (last-task accuracy $A_T$ / average accuracy $\bar{A}$) of PTM-based analytic CIL. All fine-tuning modules are trained solely on the first task ($\mathcal{D}_1$) and subsequently frozen. The dimension of the random buffer is 8192, and the default rank for LoRA is 64. A study investigating the observed ViT-LoRA collapse is provided in *Appendix A.2*.

| Item | Backbone (Parameter Weight + Fine-tuning Strategy) | CIFAR (10 tasks) | CIFAR (20 tasks) | ImageNet-R (10 tasks) | ImageNet-R (20 tasks) |
|------|---------------------------------------------------|------------------|------------------|-----------------------|-----------------------|
| i    | ViT (ViT-B/16, Fixed)                             | 84.61/89.69      | 84.64/89.78      | 70.12/75.57           | 69.95/75.98           |
| ii   | ViT-PT (ViT-B/16 + Prompt Tuning)                 | 71.25/80.34      | 48.12/59.70      | 68.63/75.10           | 70.93/76.86           |
| iii  | ViT-LoRA (ViT-B/16 + LoRA)                        | 90.62/94.26      | 3.36/19.80       | 78.90/84.60           | 77.27/82.86           |
| iv   | ViT-Adapter (ViT-B/16 + Adapters)                 | 90.80/94.33      | 89.39/93.43      | 77.78/82.94           | 76.17/81.79           |
| v    | CLIP-LoRA (CLIP-Visual + LoRA)                    | 68.90/78.98      | 49.18/62.72      | 25.48/35.98           | 12.68/21.11           |
| vi   | CLIP-Adapter (CLIP-Visual + Adapters)             | 37.97/49.21      | 31.11/43.46      | 9.32/15.92            | 8.83/15.67            |

methods are confined to frozen feature extractors, lacking the plasticity to accommodate semantic drift in long sequences. Our work bridges this gap by calibrating the analytic process with a dual-branch VLM framework.

## 3. Preliminaries and Motivation

We consider the CIL setting where a model learns from a sequence of tasks $\mathcal{T} = \{1, \ldots, T\}$. Each task $t$ introduces a dataset $\mathcal{D}_t = \{(x_i, y_i)\}_{i=1}^{N_t}$ with inputs $x_i \in \mathcal{X}$ and labels $y_i \in \mathcal{C}_t$, satisfying $\mathcal{C}_{t_i} \cap \mathcal{C}_{t_j} = \varnothing$ for $i \neq j$. The objective is to optimize a composite function $f_\theta \circ W$, where $f_\theta$ is the feature extractor and $W$ is the classifier, to minimize the prediction error on the union of all seen classes $\mathcal{C}_{1:t}$, strictly prohibiting access to raw data from previous tasks $\mathcal{T}_{<t}$.

### 3.1. Analytic CIL protocol

Analytic learning offers a recursive, closed-form alternative to gradient-based optimization. Following the ACIL protocol (Zhuang et al., 2022), the feature extractor $f_\theta$ is typically optimized *only on the first task* $\mathcal{D}_1$ and subsequently frozen to maintain a stationary feature distribution. To enhance linear separability, a fixed random projection buffer maps backbone features $\mathbf{F} = f_\theta(x) \in \mathbb{R}^D$ to a high-dimensional space $\mathbf{F}^B = \sigma(\mathbf{F}W^B) \in \mathbb{R}^{D_B}$, where $W^B$ is a frozen Gaussian matrix and $\sigma$ is the ReLU activation.

The learning objective at task $t$ is to solve the ridge regression (Murphy, 2012) problem for the classifier $W$:

$$\min_W \|\mathbf{F}^B W - Y^{OH}\|_F^2 + \lambda\|W\|_F^2 \tag{1}$$

where $Y^{OH}$ denotes the one-hot label matrix. The optimal solution is given by $W = (\Phi + \lambda I)^{-1}(\mathbf{F}^B)^T Y^{OH}$, where $\Phi = (\mathbf{F}^B)^T \mathbf{F}^B$ is the autocorrelation matrix.

In the incremental setting, recursive least squares (RLS) is employed to update the inverse correlation matrix $R_t = (\Phi_t + \lambda I)^{-1}$ and classifier $W_t$ derived from task $t-1$:

$$R_t = R_{t-1} - R_{t-1}(\mathbf{F}_t^B)^T(I + \mathbf{F}_t^B R_{t-1}(\mathbf{F}_t^B)^T)^{-1}\mathbf{F}_t^B R_{t-1}$$
$$W_t = W_{t-1} + R_t(\mathbf{F}_t^B)^T(Y_t^{OH} - \mathbf{F}_t^B W_{t-1}) \tag{2}$$

**The recursive dilemma.** The validity of Eq. (2) strictly hinges on the stationarity of $f_\theta$. The matrix $R_{t-1}$ encapsulates the precise geometric structure of past data projected by $f_{\theta_{old}}$. Any update to $\theta$ (*e.g.*, fine-tuning on Task $t$) would alter the mapping of past data, rendering the stored $R_{t-1}$ invalid. We term this phenomenon *Gram Matrix Inconsistency*. This creates a dilemma: keeping $f_\theta$ frozen ensures stability but limits plasticity, while updating $f_\theta$ enables plasticity but destroys the recursive history.

### 3.2. Theoretical motivation: generalization gap in rigid representations

Under the standard protocol where $\theta$ is frozen after Task 1, performance degradation stems strictly from *representation rigidity*. We formalize this as a subspace approximation problem. Let the optimal semantic features for task $t$ lie in a subspace $\mathcal{V}_t$. The feature extractor optimized on the initial task spans a fixed subspace $\mathcal{S}_1 = \text{span}(f_{\theta_1})$. The approximation error on a future task is fundamentally constrained by the coverage of this frozen basis.

*Remark* 3.1 (**Representation Rigidity**). Let $y \in \mathcal{Y}_t$ be the target function for a future task $t$. The expected approximation error of the analytic classifier is inherently lower-bounded by the projection residual of the task's ideal semantics onto the frozen feature space $\mathcal{S}_1$:

$$\mathcal{E}_{task_t} \geq \|(I - \mathbf{P}_{\mathcal{S}_1})y\|^2 \tag{3}$$

where $\mathbf{P}_{\mathcal{S}_1}$ is the orthogonal projection operator onto the feature space learned on Task 1.

Geometrically, this projection residual reflects the alignment gap between the initial task distribution $\mathcal{P}_1$ and the future task distribution $\mathcal{P}_t$. The formal geometric justification for this lowerbound, along with a detailed analysis of its correlation with task distribution distance (via Grassmann distance), is provided in *Appendix A.1*. This formalization suggests that initial optimization on $\mathcal{P}_1$ aggressively rotates the feature manifold $\mathcal{S}_1$ to minimize empirical risk, rendering the learned features orthogonal to the semantic directions required for novel classes (Kumar et al., 2022; Kirichenko et al., 2023). Consequently, the representation space suffers from a "semantic blind spot", creating a theoretical impera-

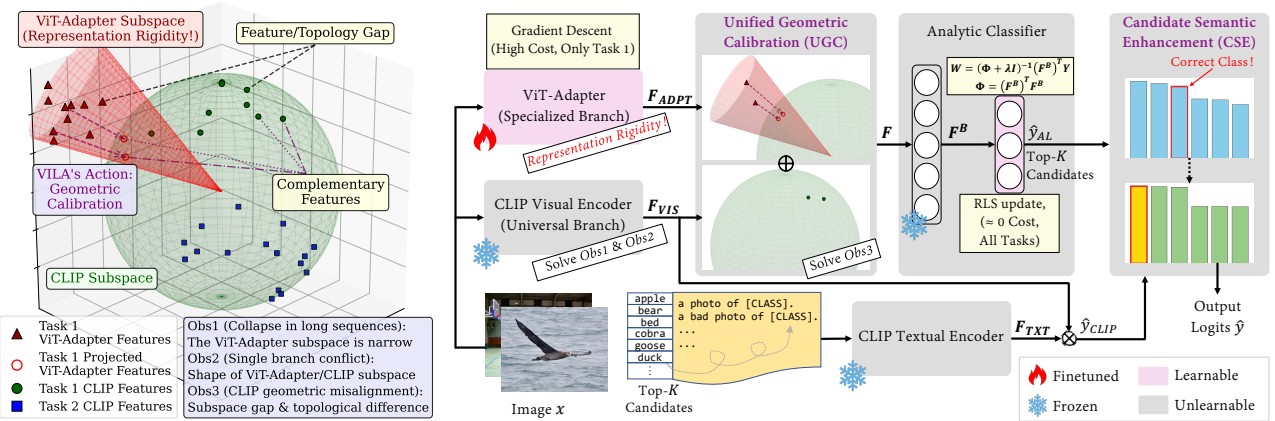

*Figure 2.* From observation to solution. **Left**: feature space trilemma. The specialized ViT-Adapter branch collapses into a rigid subspace, which creates a geometric misalignment with the universal CLIP hypersphere and fails to cover future-task features. **Right**: VILA asymmetric dual-branch architecture. It integrates a frozen universal branch to mitigate stability-plasticity dilemma (*Obs1 & Obs2*). The UGC module projects heterogeneous features onto a unified manifold to address misalignment (*Obs3*). The CSE module leverages text priors to rectify decision boundaries. Solely training on Task 1 and updating classifier via analytic learning ensures extreme efficiency.

tive for incorporating a complementary, invariant subspace.

### 3.3. Empirical verification: the plasticity-stability-geometry trilemma

To rigorously verify the theoretical bound derived in **Remark 3.1**, we conducted an extensive series of pilot studies. For clarity and focus, we curate and present the most representative results in Table 1.

*Obs1: Specialization on the initial task may lead to feature collapse in long sequences.* Consistent with our rigidity analysis, *ViT-LoRA* (iii) exhibits a stark contrast. While achieving superior accuracy on short sequences (90.62% on CIFAR 10 tasks), performance collapses on longer sequences (3.36% at 20 tasks). This confirms that optimizing exclusively for $\mathcal{D}_1$ may collapse the feature rank, rendering the specialized subspace $\mathcal{S}_1$ incapable of spanning the semantic dimensions for distant future tasks. A similar representation collapse is visualized by the red distribution in Figure 2 (Left).

*Obs2: A single branch cannot simultaneously satisfy the stability-plasticity dilemma.* Comparing *ViT* (i) and *ViT-LoRA* (iii) reveals a fundamental conflict. The former maintains stability via its universal basis but lacks discriminability, while the latter gains initial plasticity but sacrifices OOD generalization. This validates that a single branch cannot simultaneously minimize the projection residual for both current and future tasks. This inherent conflict between universal stability and task-specific plasticity is illustrated by the topological divergence between distributions in Figure 2 (Left).

*Obs3: Naive initialization with CLIP weights fails due to*

*intrinsic geometric misalignment.* Crucially, simply initializing with CLIP's visual weights (v, vi) fails catastrophically in the analytic setting (*e.g.*, 12.68% on ImageNet-R). This holds true even with standard preprocessing ($\ell_2$-normalization). We attribute this to *intrinsic geometric misalignment*: CLIP's features are optimized for contrastive matching on a specific manifold, which is structurally incompatible with the RLS solver's assumption of Euclidean linear separability (Eq. 1). As shown in Figure 2 (Left), there is an obvious gap and topological difference between the specialized and universal subspaces. Consequently, neither direct projection nor simple normalization suffices to bridge this gap and construct a valid decision boundary.

The theoretical analysis and empirical evidence converge on a single conclusion: to break the representation rigidity bottleneck, we must design a system that (1) retains the *discriminative plasticity* of a specialized adapter, (2) anchors it with the *generalizable stability* of a universal baseline, and (3) *geometrically aligns* these heterogeneous subspaces. This motivates our VILA framework.

## 4. The VILA Framework: Bridging Specialized and Universal Subspaces

Motivated by the geometric bottlenecks identified above, we propose VILA, an asymmetric dual-branch framework designed to construct an ideal hypothesis space $\mathcal{S}_{total} = \mathcal{S}_{spec} \oplus \mathcal{S}_{uni}$. Here, $\mathcal{S}_{spec}$ represents the specialized subspace that minimizes bias on the initial task distribution, while $\mathcal{S}_{uni}$ denotes the universal invariant subspace that bounds the projection residual for future tasks. The overall workflow is illustrated in Figure 2 (Right).

## 4.1. Asymmetric dual-branch architecture

The efficacy of VILA hinges on selecting appropriate backbone configurations to instantiate these two theoretical subspaces. We ground our architectural choices directly in the performance regimes observed in our pilot study (Sec 3.3).

**Adapter-based specialized branch.** We require a plastic component to capture the nuances of the initial task $\mathcal{D}_1$. While various PEFT methods are candidates, our pilot study reveals a critical performance divergence. Despite its popularity, *ViT-LoRA* (iii) exhibits severe feature collapse on long sequences (CIFAR 20 tasks). We hypothesize that the strict low-rank constraint limits its adaptation capacity, resulting in underfitting on $\mathcal{D}_1$. This failure yields a subspace that is inadequate for both current and future tasks. In contrast, *ViT-Adapter* (iv) maintain remarkable stability (89.39% at 20 tasks). This suggests that the bottleneck architecture better preserves the backbone's pre-trained topology while offering sufficient plasticity. Consequently, we select the ViT-Adapter ($f_{ADPT}$) as our specialized branch, trained on $\mathcal{D}_1$ and subsequently frozen.

**CLIP-based universal branch.** To mitigate the "semantic blind spots" (null space) of the specialized branch, a secondary branch is required. However, training another ViT on the same data risks representational homogenization, converging to the same collapsed subspace. Therefore, we strategically employ a frozen CLIP as the universal anchor. Crucially, as shown in Table 1 (v, vi), CLIP features alone perform poorly in analytic classification. This indicates that while CLIP provides a rich "universal basis", its features are not geometrically aligned for direct linear readout by the RLS solver. This observation necessitates the specific integration modules proposed below.

## 4.2. Unified geometric calibration (UGC)

Our first objective is to coherently merge the two branches into a unified feature vector $\mathbf{F}$. However, a direct concatenation faces a critical geometric misalignment. As analyzed in *Obs3* (Sec 3.3), CLIP features intrinsically reside on a hypersphere optimized for cosine similarity. In contrast, the supervised ViT-Adapter features occupy a high-magnitude unbounded Euclidean space.

We introduce a feature-level calibration to harmonize these heterogeneous representations. Given an input image $x$, we extract the universal features $\mathbf{F}_{VIS} = f_{CLIP}^{vis}(x) \in \mathbb{R}^{D_C}$ and the specialized features $\mathbf{F}_{ADPT} = f_{ADPT}(x) \in \mathbb{R}^D$. UGC projects the unbounded Adapter features onto the same hyperspherical manifold as CLIP via $\ell_2$-normalization. This calibration is essential to prevent the high-magnitude supervised features from numerically dominating the RLS update (Eq. 2), which would otherwise marginalize the contribution of the universal branch. The fused representation

$\mathbf{F}$ is defined as:

$$\mathbf{F} = \left[ \frac{\mathbf{F}_{ADPT}}{\|\mathbf{F}_{ADPT}\|_2} ; \frac{\mathbf{F}_{VIS}}{\|\mathbf{F}_{VIS}\|_2} \right] \in \mathbb{R}^{D+D_C} \quad (4)$$

This calibrated concatenation physically realizes the direct sum $\mathcal{S}_{spec} \oplus \mathcal{S}_{uni}$. The aligned features are then projected via the random buffer $W^B$ to $\mathbf{F}^B = \sigma(\mathbf{F}W^B)$, serving as the robust input for the analytic classifier.

## 4.3. Candidate semantic enhancement (CSE)

While the analytic classifier produces a robust primary prediction $\hat{y}_{AL} = \mathbf{F}^B W$, it fundamentally operates on a linear decision boundary within the random projection space. As the task sequence elongates, complex inter-class relationships (*e.g.*, fine-grained species) may become linearly inseparable. To address this, we introduce a decision-level refinement that leverages the zero-shot reasoning capability of the VLM to rectify potential misclassifications.

**Efficient candidate filtering.** Directly matching the image against all accumulated class texts is computationally expensive and prone to false positives from irrelevant classes. Instead, we utilize the analytic prediction $\hat{y}_{AL}$ as a high-quality proposal generator. We extract the indices of the top-$K$ logits to form a candidate set $\mathcal{K} = \text{TopK}(\hat{y}_{AL}, K) \subset \mathcal{C}_{1:t}$. This ensures that semantic verification is only performed on the most plausible classes, significantly reducing inference latency.

**Sparse semantic refinement.** For each class $c \in \mathcal{K}$, we construct a robust textual prototype $\mathbf{F}_{TXT}^c$. To mitigate sensitivity to linguistic phrasing, we employ a prompt ensemble strategy $\mathcal{P}$, averaging embeddings from diverse templates (see *Appendix B.1*) to obtain the final prototype:

$$\mathbf{F}_{TXT}^c = \frac{1}{|\mathcal{P}|} \sum_{p \in \mathcal{P}} f_{CLIP}^{txt}(p(c)) \quad (5)$$

The visual embedding $\mathbf{F}_{VIS}$ is then matched against these specific prototypes. The refinement score vector $\hat{y}_{CLIP} \in \mathbb{R}^{|\mathcal{C}_{1:t}|}$ is computed sparsely:

$$\hat{y}_{CLIP}[c] = \begin{cases} \langle \frac{\mathbf{F}_{VIS}}{\|\mathbf{F}_{VIS}\|_2}, \frac{\mathbf{F}_{TXT}^c}{\|\mathbf{F}_{TXT}^c\|_2} \rangle & \text{if } c \in \mathcal{K} \\ 0 & \text{otherwise} \end{cases} \quad (6)$$

The final prediction is obtained by fusing the analytic posterior and the semantic prior: $\hat{y} = \hat{y}_{AL} + \hat{y}_{CLIP}$. This ensures a synergistic balance: the analytic solver handles the bulk of discrimination efficiently, while CLIP provides fine-grained semantic verification for the most ambiguous boundaries.

**Algorithmic summary.** The complete pipeline of VILA is formalized in Algorithm 1 (learning phase) and Algorithm 2 (inference phase). By explicitly decoupling the initial optimization (gradient descent) from the continuous

---

**Algorithm 1** Learning phase of VILA

---

**input** Task stream $\mathcal{D}_1, \ldots, \mathcal{D}_T$; ViT-Adapter $f_{ADPT}$; Frozen CLIP $f_{CLIP}^{vis}$; Buffer $W^B$; Reg. $\lambda$.
**output** Fintuned ViT-Adapter $f_{ADPT}$; Continually learned analytic classifier $(W, R)$.
 1: Initialize $W \leftarrow \mathbf{0}$, $R \leftarrow \lambda^{-1}I$.
 2: Optimize adapter parameters in $f_{ADPT}$ on $\mathcal{D}_1$ via SGD.
 3: Freeze $f_{ADPT}$ permanently.
 4: **for** each task $t$ with dataset $\mathcal{D}_t$ **do**
 5:     **for** batch $(x, y) \in \mathcal{D}_t$ **do**
 6:         $\mathbf{F}_{ADPT} \leftarrow f_{ADPT}(x)$;   $\mathbf{F}_{VIS} \leftarrow f_{CLIP}^{vis}(x)$
 7:         $\mathbf{F} \leftarrow \left[ \frac{\mathbf{F}_{ADPT}}{\|\mathbf{F}_{ADPT}\|_2} ; \frac{\mathbf{F}_{VIS}}{\|\mathbf{F}_{VIS}\|_2} \right]$
 8:         $\mathbf{F}^B \leftarrow \sigma(\mathbf{F}W^B)$
 9:         Update $R$ and $W$ using $\mathbf{F}^B, y$ via RLS (Eq. 2)
10:     **end for**
11: **end for**

---

**Algorithm 2** Inference phase of VILA

---

**input** Test image $x$; Models $f_{ADPT}, f_{CLIP}$; $W^B$; Classifier $W$; Candidate $K$.
**output** Predicted class label $\hat{y}$.
 1: Extract calibrated feature $\mathbf{F}$ via UGC (Eq. 4).
 2: Compute projection $\mathbf{F}^B \leftarrow \sigma(\mathbf{F}W^B)$.
 3: $\hat{y}_{AL} \leftarrow \mathbf{F}^B W$
 4: $\mathcal{K} \leftarrow \text{TopK}(\hat{y}_{AL}, K)$ ▷ Select top-K candidate indices
 5: **for** each class $c \in \mathcal{K}$ **do**
 6:     Construct prototype $\mathbf{F}_{TXT}^c$ via Prompt Ensemble (Eq. 5)
 7:     Compute refinement score $\hat{y}_{CLIP}$ via Eq. 6
 8: **end for**
 9: $\hat{y} \leftarrow (\hat{y}_{AL} + \hat{y}_{CLIP})$

---

analytic updates (recursive RLS), VILA ensures that the backpropagation overhead is incurred only once during its entire lifecycle. Further details of VILA algorithm and its efficient deployment variant using offline semantic caching (VILA-OSC) are provided in *Appendix B.2*.

# 5. Experiments

## 5.1. Experimental setup

**Datasets.** To verify the robustness of VILA across diverse visual domains and granularity levels, we conduct comprehensive evaluations on 8 benchmarks: (1) *Coarse-grained object recognition*: CIFAR100 (Krizhevsky et al., 2009) and ImageNet-R (Hendrycks et al., 2021); (2) *Fine-grained classification*: CUB200 (Wah et al., 2011) (Birds), FGVC-Aircraft (Maji et al., 2013), StanfordCars (Krause et al., 2013), and Food101 (Bossard et al., 2014); (3) *Scene & action recognition*: UCF101 (Soomro et al., 2012) and SUN397 (Xiao et al., 2010). We select a subset of classes:

100 classes for CIFAR, Aircraft, Cars, Food, and UCF; 200 classes for ImageNet-R and CUB; 300 classes for SUN. All classes are randomly shuffled.

**Implementation details.** We construct the specialized branch using a ViT-B/16 (Dosovitskiy, 2020) backbone equipped with the Adapters, and the universal branch using a frozen CLIP-ViT-B/16 (LAION-400M pre-trained) (Ilharco et al., 2021). The feature dimensions are $D = 768$ and $D_C = 512$, respectively. We train the ViT-Adapter on the first task using the AdamW optimizer with an initial learning rate of 0.01, decaying via a cosine annealing schedule. The random buffer dimension is set to $D_B = 16384$. For the regularization hyperparameter $\lambda$, we employ a self-adaptive selection strategy. We perform leave-one-out cross-validation (LOOCV) on the base task to automatically select the optimal $\lambda$ from the candidate set $\{10^{-8}, 10^{-7}, \ldots, 10^0\}$. All experiments are implemented in PyTorch and conducted on a single NVIDIA RTX 4090 GPU. We report results over 5 distinct global random seeds: $\{1993, 1, 56, 254, 602\}$.

**Baselines.** We benchmark VILA against a comprehensive suite of SOTA methods categorized by their adaptation mechanism: (1) *Prompt-based*: L2P (Wang et al., 2022b), DualPrompt (Wang et al., 2022a), CODA-Prompt (Smith et al., 2023); (2) *Adapter-based*: APER-Adapter (Zhou et al., 2025a), TUNA (Wang et al., 2025b); (3) *LoRA-based*: SD-LoRA (Wu et al., 2025), CL-LoRA (He et al., 2025); (4) *VLM-based*: ENGINE (Zhou et al., 2025b); (5) *Analytic-based*: RanPAC (McDonnell et al., 2023), LoRanPAC (Peng et al., 2025).

**Evaluation Metrics.** We report two standard metrics: (1) Average accuracy $\bar{A} = \frac{1}{T} \sum_{t=1}^{T} Acc_t$: The mean of test accuracies across all incremental stages, where $Acc_t$ is the accuracy on all seen classes $\mathcal{C}_{1:t}$ after learning task $t$. (2) Last-task accuracy $A_T$: The final accuracy on all classes after the entire sequence is learned, reflecting the ultimate plasticity-stability trade-off.

## 5.2. Benchmark comparison

**Quantitative superiority.** Table 2 presents a comprehensive comparison against SOTA PTM-based methods. VILA sets a new standard for analytic learning, outperforming existing approaches by substantial margins of **5.42%** in $A_T$ and **4.75%** in $\bar{A}$. This stems from three factors: (1) *Fine-grained adaptation.* The specialized branch mitigates representation rigidity to capture subtle inter-class distinctions (*e.g.*, exceeding LoRanPAC by +20.11% on Cars and +6.24% on Aircraft). (2) *OOD robustness.* The geometric calibration aligns heterogeneous subspaces to enable robust generalization under domain shifts (*e.g.*, exceeding ENGINE by +4.17% on ImageNet-R). (3) *Long-term stability.* The frozen universal anchor circumvents parameter drift to prevent catastrophic forgetting in long sequences (*e.g.*,

*Table 2.* Comparison of last-task accuracy $A_T$ (%) and average incremental accuracy $\bar{A}$ (%) among PTM-based methods across diverse benchmarks. $T$ denotes the total number of tasks. The best and second-best results are highlighted in **bold** and blue, respectively.

| Metric | Dataset | CIFAR | | ImageNet-R | | CUB | UCF | Aircraft | | Cars | Food | SUN | | Avg. |
|---|---|---|---|---|---|---|---|---|---|---|---|---|---|---|
| | $T =$ | 10 | 20 | 20 | 40 | 20 | 20 | 10 | 20 | 20 | 20 | 30 | 50 | |
| $A_T$ | L2P | 84.29 | 77.70 | 70.13 | 65.07 | 64.21 | 72.41 | 28.74 | 18.90 | 20.52 | 70.03 | 57.41 | 54.98 | 57.03 |
| | DualPrompt | 82.30 | 77.23 | 65.98 | 60.13 | 65.95 | 76.17 | 25.50 | 10.14 | 18.66 | 69.22 | 62.23 | 62.03 | 56.29 |
| | CODA-Prompt | 86.85 | 79.59 | 70.33 | 61.02 | 68.45 | 76.09 | 27.24 | 15.54 | 15.31 | 72.32 | 64.88 | 58.49 | 58.01 |
| | APER-Adapter | 85.80 | 82.77 | 70.52 | 66.70 | 84.68 | 86.66 | 30.96 | 30.93 | 41.69 | 77.98 | 72.51 | 72.46 | 66.97 |
| | TUNA | 91.09 | 90.42 | 77.33 | 75.55 | 88.46 | 94.51 | 54.49 | 47.52 | 54.05 | 79.40 | 75.09 | 75.30 | 75.27 |
| | SD-LoRA | 87.50 | 82.28 | 71.20 | 63.27 | 70.46 | 73.29 | 40.29 | 23.76 | 35.49 | 42.91 | 54.06 | 39.21 | 56.98 |
| | CL-LoRA | 88.17 | 84.91 | 78.40 | 73.50 | 73.92 | 79.23 | 45.57 | 26.76 | 40.46 | 76.42 | 71.34 | 69.27 | 67.33 |
| | ENGINE | 79.29 | 79.10 | 80.47 | 80.03 | 78.54 | 87.91 | 58.69 | 55.51 | 89.49 | 83.88 | 78.16 | 77.99 | 77.42 |
| | RanPAC | 86.40 | 86.36 | 71.83 | 71.70 | 88.93 | 97.73 | 59.23 | 58.33 | 66.81 | 84.56 | 79.65 | 79.75 | 77.61 |
| | LoRanPAC | 91.32 | 90.34 | 74.95 | 73.42 | **90.25** | 98.22 | 59.38 | 59.32 | 70.17 | 86.84 | 80.77 | 80.76 | 79.64 |
| | VILA (Ours) | **91.94** | **91.18** | **85.28** | **84.20** | 89.36 | 98.71 | 68.02 | 65.56 | 90.28 | **88.89** | 83.73 | 83.55 | **85.06** |
| $\bar{A}$ | L2P | 89.23 | 84.20 | 75.86 | 72.04 | 75.93 | 83.58 | 41.79 | 32.32 | 29.92 | 77.62 | 68.54 | 66.45 | 66.46 |
| | DualPrompt | 87.36 | 84.85 | 72.53 | 68.23 | 76.44 | 86.05 | 40.12 | 23.57 | 31.05 | 79.58 | 73.64 | 73.17 | 66.38 |
| | CODA-Prompt | 91.30 | 86.17 | 75.93 | 66.48 | 76.57 | 84.87 | 30.50 | 19.44 | 33.08 | 81.39 | 76.95 | 72.07 | 66.23 |
| | APER-Adapter | 90.91 | 88.49 | 77.20 | 74.02 | 91.29 | 91.18 | 39.92 | 41.36 | 55.62 | 85.14 | 80.33 | 80.43 | 74.66 |
| | TUNA | 94.60 | 94.38 | 82.59 | 80.77 | 93.18 | 97.49 | 63.18 | 58.75 | 69.05 | 88.11 | 83.92 | 84.32 | 82.53 |
| | SD-LoRA | 92.31 | 88.89 | 73.82 | 65.05 | 76.87 | 84.29 | 57.29 | 39.22 | 48.28 | 21.40 | 35.45 | 25.52 | 59.03 |
| | CL-LoRA | 92.41 | 90.87 | 84.83 | 80.63 | 82.82 | 89.25 | 59.71 | 43.77 | 56.43 | 84.47 | 79.66 | 77.74 | 76.88 |
| | ENGINE | 86.87 | 87.12 | 86.21 | 85.98 | 86.32 | 92.97 | 69.69 | 68.79 | 94.03 | 89.86 | 85.04 | 85.09 | 84.83 |
| | RanPAC | 90.93 | 91.27 | 77.71 | 77.32 | 92.97 | 98.83 | 68.10 | 68.29 | 67.77 | 90.05 | 86.04 | 86.12 | 82.95 |
| | LoRanPAC | 94.60 | 93.90 | 79.90 | 79.02 | **93.48** | 99.19 | 67.99 | 69.27 | 80.04 | 91.70 | 86.74 | 86.89 | 85.23 |
| | VILA (Ours) | **95.09** | **94.72** | **89.81** | **89.16** | 93.16 | 99.35 | 76.26 | 75.74 | 94.19 | 93.32 | 89.43 | 89.53 | **89.98** |

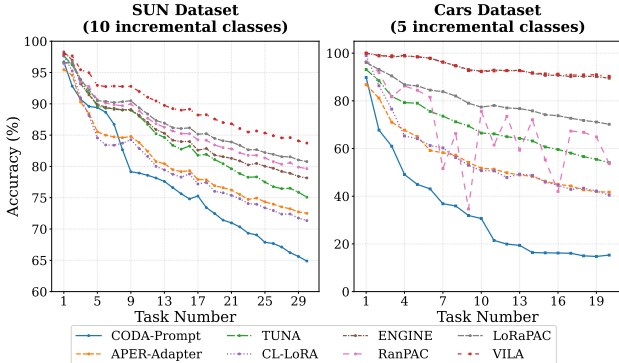

*Figure 3.* Incremental performance trajectory. Comparison of step-wise accuracy on SUN (Left, $T = 30$) and Cars (Right, $T = 20$). VILA demonstrates superior stability and resistance to forgetting compared to SOTA methods.

*Table 3.* Comparison under strict online learning setting (1 epoch).

| Method | ImageNet-R ($T = 10$) | | CUB ($T = 20$) | |
|---|---|---|---|---|
| | $A_T$ | $\bar{A}$ | $A_T$ | $\bar{A}$ |
| L2P | $64.57_{\pm0.46}$ | $70.11_{\pm0.80}$ | $50.13_{\pm1.41}$ | $65.08_{\pm1.36}$ |
| DualPrompt | $61.48_{\pm0.35}$ | $67.66_{\pm0.77}$ | $47.94_{\pm2.12}$ | $61.83_{\pm3.21}$ |
| CODA-Prompt | $58.23_{\pm0.53}$ | $62.96_{\pm0.88}$ | $30.66_{\pm1.32}$ | $42.38_{\pm2.00}$ |
| APER-Adapter | $65.96_{\pm4.81}$ | $69.65_{\pm1.42}$ | $85.22_{\pm0.02}$ | $90.39_{\pm0.81}$ |
| TUNA | $79.05_{\pm0.50}$ | $84.16_{\pm0.67}$ | $88.32_{\pm0.21}$ | $92.68_{\pm0.78}$ |
| SD-LoRA | $67.27_{\pm4.20}$ | $70.26_{\pm9.08}$ | $46.41_{\pm2.99}$ | $65.29_{\pm1.63}$ |
| CL-LoRA | $72.17_{\pm0.63}$ | $77.52_{\pm0.90}$ | $80.66_{\pm0.87}$ | $88.06_{\pm0.93}$ |
| ENGINE | $79.53_{\pm0.18}$ | $85.67_{\pm0.66}$ | $59.12_{\pm0.26}$ | $77.12_{\pm1.32}$ |
| RanPAC | $71.75_{\pm0.30}$ | $77.17_{\pm0.80}$ | $88.92_{\pm0.39}$ | $92.84_{\pm0.80}$ |
| LoRanPAC | $75.23_{\pm0.82}$ | $79.72_{\pm0.94}$ | **$90.26_{\pm0.25}$** | $93.53_{\pm0.68}$ |
| VILA | **$84.52_{\pm0.27}$** | **$88.97_{\pm0.45}$** | $89.37_{\pm0.36}$ | $93.23_{\pm0.58}$ |

maintaining 89.53% on SUN where others often collapse).

**Sequential stability.** Figure 3 visualizes the step-wise trajectory. On the Cars dataset, VILA maintains a smooth curve, effectively insulating the model from the severe volatility observed in RanPAC. On SUN dataset, it exhibits the slowest decay rate, surpassing the nearest competitor by ∼3.0% at the final task. These results confirm that the universal branch serves as a stable semantic anchor, preventing decision boundary drift over long sequences. Complete incremental performance curves for all remaining benchmarks are detailed in *Appendix C.1*.

**One-pass online learning.** In the strict 1-epoch setting (Ta-

ble 3), VILA exhibits minimal degradation and consistently outperforms SOTA baselines (*e.g.*, +4.99% on ImageNet-R). Unlike gradient-based methods that suffer from under-fitting due to insufficient updates, VILA's recursive analytic solver derives mathematically optimal solutions instantaneously, ensuring superior data efficiency.

**Efficiency trade-off.** Figure 4 places VILA on the optimal *Pareto frontier*. VILA requires only ∼7 mins for training and ∼7.5 mins for inference. This strikes a superior balance between latency and accuracy, avoiding the computational overhead of heavy adapters or iterative optimization common in prior arts. Detailed efficiency breakdowns per dataset are provided in *Appendix C.2*.

**Computational complexity.** To ensure our gains are not driven by unfair parameter or FLOP scaling, we compare

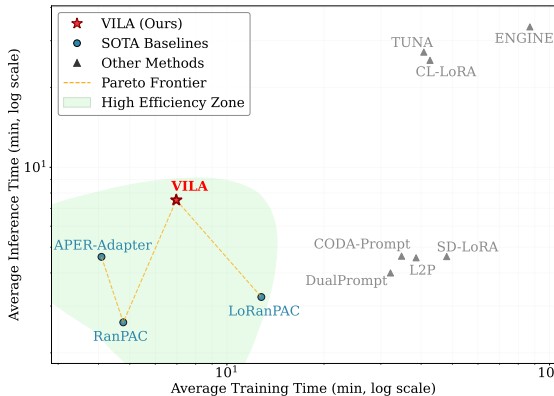

*Figure 4.* Comparison of total training vs. inference time averaged over 8 datasets ($T = 20$ for all). VILA occupies the high efficiency zone, offering the best trade-off by being significantly faster than complex baselines while outperforming lighter methods.

*Table 4.* Comparison of computational complexity (FLOPs and Parameters) across various SOTA PTM-based methods. Experiment uses UCF $T = 10$ and pytorch third-party *thop* to calculate FLOPs. For VILA using offline semantic caching, the text encoder is offloaded, significantly reducing the active inference overhead.

| Method | FLOPs (G) | Params (M) |
|---|---|---|
| L2P | 35.85 | 171.82 |
| DualPrompt | 33.73 | 172.00 |
| CODA-P | 33.73 | 89.72 |
| APER | 33.96 | 172.93 |
| TUNA | 169.21 | 89.29 |
| SD-LoRA | 16.94 | 172.04 |
| CL-LoRA | 167.99 | 90.25 |
| ENGINE | 299.57 | 144.97 |
| RanPAC | 17.10 | 87.79 |
| LoRanPAC | 17.10 | 86.84 |
| VILA | 493.47 | 259.06 |
|  | 33.98 | 174.27 |

the computational complexity of VILA and its practical deployment variant (detailed in *Appendix B.2*) against SOTA methods in Table 4. While the standard VILA utilizes both visual and textual branches online, its deployment variant explicitly offloads the heavy text encoder. This strategy constrains the active online inference cost to 33.98G FLOPs and 174.27M parameters.

As observed in Table 4, baselines such as RanPAC and LoRanPAC inherently operate with notably smaller parameter counts and FLOPs in their default configurations. To rigorously address this disparity, we conduct extended experiments in *Appendix C.3* under a strictly matched computational budget, confirming that VILA still demonstrates overwhelming superiority even when the baselines are scaled to an equivalent or larger capacity.

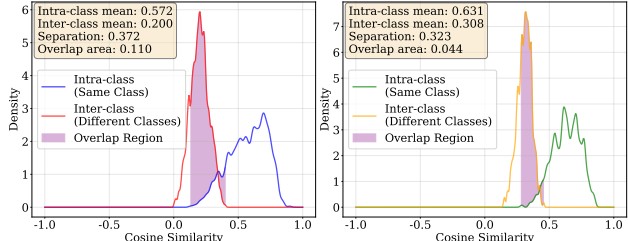

*Figure 5.* Comparison to the baseline (left) and with UGC (right) on ImageNet-R. We visualize the density estimation of cosine similarities for intra-class (green/blue) and inter-class (orange/red) pairs. The shaded area (colored in purple) quantifies the confusion region.

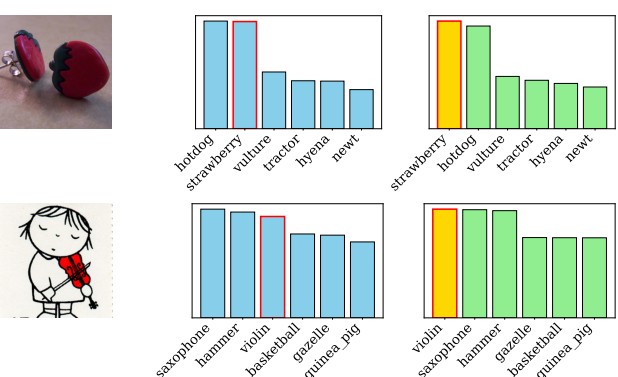

*Figure 6.* Correction on hard samples on ImageNet-R. Blue bars show baseline misclassifications (w/o CSE). Green/Yellow bars show VILA rectifies these hallucinations via semantic context (w CSE).

### 5.3. Case Study

**Geometric calibration (UGC).** Figure 5 quantifies discriminability. The baseline suffers from severe intra/inter-class overlap (0.110), causing confusion. VILA aggressively compresses this overlap by ~60% (to 0.044) and shifts intra-class coherence towards 1.0. This confirms UGC acts as a geometric regularizer, effectively sharpening decision boundaries.

**Semantic correction (CSE).** Figure 6 demonstrates robustness against visual-semantic mismatches. While the baseline misclassifies deceptive samples (*e.g.*, *"strawberry earring"* → *"hotdog"*) due to texture bias, CSE successfully rectifies these hallucinations. By leveraging universal semantics, VILA suppresses low-level visual noise and enforces semantically consistent predictions.

### 5.4. Ablation Study

**Core component analysis.** Table 5 validates each module's contribution. The dual-branch (DB) architecture forms the critical backbone, yielding the largest gain. Building on this, UGC further boosts performance by aligning heterogeneous

*Table 5.* Component ablation study. DB is dual-branch framework.

| DB | UGC | CSE | Aircraft ($T=10$) | | Cars ($T=10$) | |
|---|---|---|---|---|---|---|
| | | | $A_T$ | $\bar{A}$ | $A_T$ | $\bar{A}$ |
| | | | $59.12_{\pm0.57}$ | $68.99_{\pm1.54}$ | $68.27_{\pm1.06}$ | $77.70_{\pm1.39}$ |
| ✔ | | | $65.28_{\pm0.64}$ | $74.83_{\pm1.45}$ | $84.13_{\pm1.00}$ | $90.43_{\pm0.51}$ |
| ✔ | ✔ | | $67.24_{\pm0.35}$ | $76.53_{\pm1.37}$ | $90.45_{\pm0.36}$ | $94.33_{\pm0.70}$ |
| | | ✔ | $59.87_{\pm0.52}$ | $69.52_{\pm1.43}$ | $73.43_{\pm0.76}$ | $81.51_{\pm0.89}$ |
| ✔ | | ✔ | $65.40_{\pm0.51}$ | $75.00_{\pm1.46}$ | $85.18_{\pm0.95}$ | $90.98_{\pm0.58}$ |
| ✔ | ✔ | ✔ | $\mathbf{67.26_{\pm0.46}}$ | $\mathbf{76.60_{\pm1.34}}$ | $\mathbf{90.58_{\pm0.40}}$ | $\mathbf{94.47_{\pm0.71}}$ |

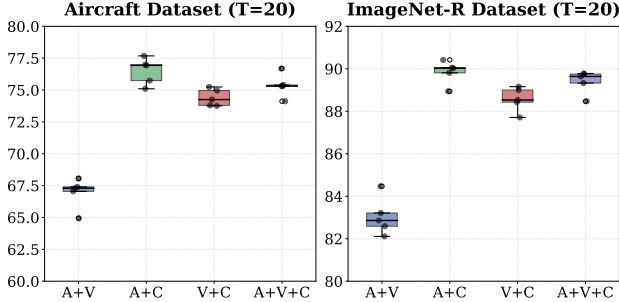

*Figure 7.* Paradigm integration ablation study ($\bar{A}$). "V" is frozen ViT, "C" is frozen CLIP, and "A" is ViT-Adapter.

*Table 6.* Backbone robustness and data leakage analysis. VILA maintains exceptional performance even when the CLIP visual encoder is replaced with a frozen DINOv2 (ViT-Base), confirming that the gains are driven by the structural innovation rather than specific pre-trained manifolds.

| Variant | VILA (DB+UGC, DINOv2) | VILA (DB+UGC, CLIP) |
|---|---|---|
| CIFAR ($T=10$) | 92.61 / 95.55 | 91.87 / 95.01 |
| CIFAR ($T=20$) | 91.54 / 95.05 | 91.09 / 94.50 |
| ImageNet-R ($T=20$) | 83.62 / 88.02 | 82.78 / 88.32 |
| ImageNet-R ($T=40$) | 81.93 / 87.13 | 82.62 / 88.36 |
| CUB ($T=20$) | 91.06 / 94.29 | 89.19 / 93.05 |
| UCF ($T=20$) | 98.69 / 99.30 | 98.60 / 99.29 |
| Aircraft ($T=10$) | 75.46 / 83.75 | 67.24 / 76.53 |
| Aircraft ($T=20$) | 74.47 / 84.01 | 61.96 / 72.44 |
| Cars ($T=20$) | 90.64 / 94.45 | 89.35 / 93.87 |
| Food ($T=20$) | 90.68 / 94.45 | 88.25 / 93.26 |
| SUN ($T=30$) | 81.65 / 87.61 | 83.60 / 89.32 |
| SUN ($T=50$) | 82.11 / 87.90 | 83.44 / 89.40 |

subspaces, while CSE provides complementary semantic refinement. Their combination achieves optimal performance, confirming the synergy of geometric constraints and semantic correction. Comprehensive ablation results for all remaining benchmarks are provided in *Appendix D.1*.

**Paradigm Integration.** Figure 7 compares feature sources. The combination of learnable Adapters with frozen CLIP features ($A+C$) consistently outperforms the ViT-based counterpart ($A+V$), highlighting the superiority of CLIP's open-world priors for CIL. Notably, fusing all features ($A+V+C$) leads to slight drops due to representational redundancy.

**Backbone capacity and data leakage.** A potential concern when utilizing CLIP features is whether the performance gains are merely a byproduct of CLIP's massive capacity or potential data leakage during its pre-training phase. To rigorously rule out this possibility, we substitute the frozen CLIP visual encoder with DINOv2 (ViT-Base) (Oquab et al., 2024). As shown in Table 6, even without explicit semantic priors, the ablated VILA variant (DB+UGC utilizing DINOv2) maintains exceptional robustness. For example, it achieves 91.54%/95.05% on CIFAR $T=20$ and 83.62%/88.02% on ImageNet-R $T=20$. Furthermore, existing methods ENGINE utilizing CLIP features fails to achieve comparable results on our benchmarks.

This powerfully proves that VILA is driven entirely by architectural innovation rather than passively relying on the specific pre-trained manifold of CLIP, *i.e.*, safely unlocking structural priors through dual-branch geometric calibration.

**Sensitivity to $K$.** Evaluating the candidate size $K$ within the CSE module shows that performance saturates rapidly at $K=5$. Accuracy remains highly stable ($<0.1\%$ fluctuation) thereafter (see *Appendix D.2*).

## 6. Conclusion and Discussion

In this paper, we propose VILA, a dual-branch analytic framework that harmonizes task-specific plasticity with universal semantic stability via a two-level calibration strategy. Extensive experiments demonstrate that VILA establishes a new Pareto frontier in efficiency and accuracy, particularly for fine-grained and long-sequence CIL. Ultimately, this work suggests that analytic learning offers a promising and resource-efficient alternative to gradient-based paradigms for adapting foundation models.

**Limitations and Future Work.** Despite these advances, three limitations remain for future exploration. *First*, the dual-branch architecture inherently doubles the inference parameter count. Future work could explore knowledge distillation to condense the model for resource-constrained edge deployment. *Second*, the memory consumption of the analytic solver grows quadratically $O(D_B^2)$ with the feature dimension, posing a bottleneck for extremely high-dimensional representations that may require low-rank approximations. *Third*, freezing the specialized backbone after the initial task assumes a consistent domain distribution. While most existing methods also struggle in such heterogeneous settings, this strategy may be suboptimal for drastic mixed-granularity streams. Investigating adaptive mechanisms may better handle these domain shifts.

## Acknowledgements

This research/project is supported by the National Research Foundation, Singapore under its National Large Language Models Funding Initiative (AISG Award No: AISG-NMLP-2024-003). Any opinions, findings and conclusions or recommendations expressed in this material are those of the author(s) and do not reflect the views of National Research Foundation, Singapore.

## Impact Statement

This paper presents VILA, a framework designed to advance the efficiency and stability of Class-Incremental Learning (CIL).

**Positive societal impact.** A primary contribution of this work is the significant reduction in computational resources required for continuous adaptation. By achieving order-of-magnitude faster training speeds and lower memory footprints compared to existing methods, VILA promotes the principles of *Green AI*, facilitating the deployment of intelligent systems on resource-constrained edge devices with reduced carbon emissions.

**Potential risks.** Our framework leverages large-scale pretrained models (*i.e.*, CLIP) to extract semantic priors. Consequently, VILA may inherit inherent biases (*e.g.*, regarding gender, race, or culture) present in the pre-training datasets of these foundation models. Users should be aware of these potential biases when deploying the system in sensitive real-world applications.

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

# A. Theoretical Justification and Motivation

## A.1. Geometric Justification of Representation Rigidity

In this section, we provide the detailed derivation for **Remark 3.1**, justifying why the analytic approximation error is inherently lower-bounded by the projection residual, and how this residual geometrically scales with the distribution shift between the initial task $\mathcal{P}_1$ and a future task $\mathcal{P}_t$.

Let $f_\theta : \mathcal{X} \to \mathbb{R}^D$ be the backbone network. After training on Task 1 ($\mathcal{P}_1$), the parameters are fixed at $\theta^*$. This frozen extractor defines a feature subspace $\mathcal{S}_1 = \text{range}(\mathbf{F}_1) \subset \mathbb{R}^D$, where $\mathbf{F}_1$ represents the feature matrix of the training data. The Analytic Classifier (AC) computes weights $W$ via Regularized Least Squares (RLS). As the regularization term $\lambda \to 0$, the AC solution converges to the orthogonal projection of the target $y$ onto $\mathcal{S}_1$.

### A.1.1. Step 1: establishing the projection lower bound

For a sample $x$ from a future task distribution $\mathcal{P}_t$, let $y \in \mathcal{Y}_t$ be the ground truth target vector. The prediction of the analytic classifier is $\hat{y} = \mathbf{P}_{\mathcal{S}_1} y$, where $\mathbf{P}_{\mathcal{S}_1}$ is the projection operator onto $\mathcal{S}_1$. By the projection theorem, the squared error norm can be decomposed as:

$$\|y - \hat{y}\|^2 = \|y - \mathbf{P}_{\mathcal{S}_1} y\|^2 = \|(I - \mathbf{P}_{\mathcal{S}_1})y\|^2 \tag{7}$$

Since any other classifier (including regularized versions) in the span of $\mathcal{S}_1$ cannot achieve a lower error than the orthogonal projection on the training set, the expected error $\mathcal{E}_{task_t}$ on the new task is strictly lower-bounded by this projection residual (the "bias" term):

$$\mathcal{E}_{task_t} \geq \mathbb{E}_{x \sim \mathcal{P}_t} \left[ \|(I - \mathbf{P}_{\mathcal{S}_1})y\|^2 \right] \tag{8}$$

This establishes the fundamental inequality presented in Eq. 3.

### A.1.2. Step 2: linking the residual to distribution distance

During the training on Task 1, maximizing the likelihood is equivalent to aligning the feature subspace $\mathcal{S}_1$ with the principal components of the Task 1 distribution. Let $\mathbf{U}_1$ be the basis of $\mathcal{S}_1$. Ideally, $\mathcal{S}_1$ captures the dominant variance of $\mathcal{P}_1$.

For the future task $t$, the target function $y$ depends on the latent semantic features inherent to distribution $\mathcal{P}_t$. Let $\mathcal{S}_t$ denote the ideal subspace that fully spans the semantics of $\mathcal{P}_t$. The target $y$ essentially lies within $\mathcal{S}_t$. The projection residual $\|(I - \mathbf{P}_{\mathcal{S}_1})y\|^2$ measures how much of the "energy" of $y$ lies outside $\mathcal{S}_1$. This is geometrically determined by the *Principal Angles* between subspaces $\mathcal{S}_1$ and $\mathcal{S}_t$.

We formalize the distribution shift as the *Grassmann Distance* (a metric for subspace distance):

$$d_G(\mathcal{S}_1, \mathcal{S}_t) = \|\sin \Theta\|_F \tag{9}$$

where $\Theta$ represents the vector of principal angles between the frozen subspace $\mathcal{S}_1$ and the target subspace $\mathcal{S}_t$. If $\mathcal{P}_1$ and $\mathcal{P}_t$ are similar (small distance), $\mathcal{S}_1 \approx \mathcal{S}_t$, and the projection captures most of $y$. Conversely, if $\mathcal{P}_t$ represents a distinct domain (*e.g.*, fine-grained vs. coarse), $\mathcal{S}_t$ creates a large angle with $\mathcal{S}_1$.

Therefore, the residual energy can be geometrically bounded by the sine of the angle between the subspaces. For a normalized target $y$, the residual scales as:

$$\|(I - \mathbf{P}_{\mathcal{S}_1})y\|^2 \approx \sin^2(\angle(\mathcal{S}_1, y)) \tag{10}$$

where $\angle(\mathcal{S}_1, y)$ is directly bounded by the Grassmann distance $d_G(\mathcal{S}_1, \mathcal{S}_t)$.

### A.1.3. Conclusion

Combining Step 1 and Step 2, we theoretically justify the geometric intuition in **Remark 3.1**: the approximation error is fundamentally bounded by the projection residual (Eq. 8), and this residual expands as the subspace divergence (Grassmann distance) between tasks increases. This confirms that representation rigidity (the inability to rotate $\mathcal{S}_1$ to match $\mathcal{P}_t$) imposes an inevitable error floor in single-branch analytic learning.

*Table 7.* Performance (last-task accuracy $A_T$ / average accuracy $\bar{A}$) of ViT-LoRA on CIFAR (20 tasks) across varying ranks.

| Rank ($r$) | 4 | 8 | 16 | 32 | 64 |
|---|---|---|---|---|---|
| $A_T$ / $\bar{A}$ | 15.95 / 26.66 | 16.43 / 27.36 | 88.91 / 93.03 | 15.97 / 26.75 | 3.36 / 19.80 |

## A.2. LoRA Rank and Hyperparameter Brittleness

In Table 1 of the main text, we observed a catastrophic performance collapse for *ViT-LoRA* on long sequences (CIFAR 20 tasks) using the default rank $r = 64$. To thoroughly address whether this collapse is an isolated hyperparameter artifact or a general structural vulnerability, we conduct an ablation study across a wider spectrum of LoRA ranks $r \in \{4, 8, 16, 32, 64\}$. The experimental setup remains identical to the pilot study (trained solely on $\mathcal{D}_1$ and subsequently frozen for analytic incremental updates).

As shown in Table 7, the representation collapse is not an anomaly but the dominant trend. The analytic CIL performance exhibits extreme *hyperparameter brittleness*. Except for a narrow "sweet spot" at $r = 16$, the performance collapses catastrophically at both lower ranks ($r \in \{4, 8\}$) and higher ranks ($r \in \{32, 64\}$).

This extreme variance underscores the fundamental limitation of single-branch architectures in non-stationary environments. In practical, open-world class-incremental learning, the length and semantic distribution of future task streams are unpredictable during the initial Task 1 training. Consequently, it is impossible to pre-determine a single optimal rank that simultaneously satisfies the stability and plasticity demands for all future scenarios. If the rank misses this narrow optimal window, the specialized subspace either suffers from severe rank deficiency or aggressive overfitting to the initial distribution. This brittleness strongly reinforces the core motivation of VILA.

# B. Algorithm and Implementation Details

## B.1. Designed Templates for Semantic Projection

To maximize the generalization capability of the universal branch, we utilize a unified set of prompt templates across all benchmarks (CIFAR-100, ImageNet-R, *etc.*). We do not tune templates for specific datasets, thereby demonstrating the intrinsic robustness of VILA. Specifically, we employ an ensemble of 19 templates designed to capture diverse visual contexts. Let {} denote the class label:

```
"itap of {}.",
"art of {}.",
"i love {}!",
"a origami {}.",
"a photo of {}.",
"a video of {}.",
"a example of {}.",
"a bad photo of {}.",
"a good photo of {}.",
"a large photo of {}.",
"a small photo of {}.",
"a old photo of {}.",
"a clean photo of {}.",
"a dirty photo of {}.",
"a blurry photo of {}.",
"a black and white photo of {}.",
"a low contrast photo of {}.",
"a high contrast photo of {}.",
"{} in a video game."
```

The selection of these specific templates is motivated by three objectives crucial for Class-Incremental Learning. By averaging the features from these diverse prompts, we obtain a **mean textual prototype** that is significantly more stable and representative than any single-prompt embedding, effectively reducing the variance caused by linguistic ambiguity.

---

**Algorithm 3** Offline semantic caching (Preparation in training phase)

---

**input** Novel class set $\mathcal{C}_t$ for the incoming task $t$; Text Encoder $f_{CLIP}^{txt}$; Current cache memory $\mathcal{M}_{txt}$.
**output** Updated lightweight prototype look-up table $\mathcal{M}_{txt}$.
 1: **for** each novel class $c \in \mathcal{C}_t$ **do**
 2:      Construct textual prompts using class name $c$ and templates $\mathcal{P}$.
 3:      Extract embeddings: $\mathbf{F}_{TXT}^c \leftarrow \frac{1}{|\mathcal{P}|} \sum_{p \in \mathcal{P}} f_{CLIP}^{txt}(p(c))$.
 4:      Update cache: $\mathcal{M}_{txt}[c] \leftarrow \mathbf{F}_{TXT}^c$.
 5: **end for**
 6: Offload $f_{CLIP}^{txt}$ from active memory to eliminate inference FLOPs.

---

**Algorithm 4** Highly efficient online prediction (Deployment in inference phase)

---

**input** Test image $x$; Models $f_{ADPT}, f_{CLIP}^{vis}$; $W^B$; Classifier $W$; Cached table $\mathcal{M}_{txt}$; Candidate size $K$.
**output** Predicted class label $\hat{y}$.
 1: Extract calibrated feature $\mathbf{F}$ and compute primary logits $\hat{y}_{AL}$ (via Algorithm 2, Lines 1-3).
 2: $\mathcal{K} \leftarrow \text{TopK}(\hat{y}_{AL}, K)$                                      ▷ Select top-K candidate indices
 3: **for** each class $c \in \mathcal{K}$ **do**
 4:      Retrieve prototype: $\mathbf{F}_{TXT}^c \leftarrow \mathcal{M}_{txt}[c]$                        ▷ O(1) offline table look-up
 5:      Compute refinement score $\hat{y}_{CLIP}$ via Eq. 6.
 6: **end for**
 7: $\hat{y} \leftarrow (\hat{y}_{AL} + \hat{y}_{CLIP})$.

---

### B.1.1. ROBUSTNESS AGAINST IMAGE DEGRADATION

We include templates such as *"a blurry photo of..."* , *"a bad photo of..."* , and *"low/high contrast..."*. These prompts explicitly instruct the text encoder to project class semantics into feature regions corresponding to corrupted or noisy inputs. This geometric alignment improves matching accuracy when the visual backbone encounters varying data quality or distribution shifts in continuous streams.

### B.1.2. ACCOMMODATION OF DOMAIN SHIFTS

To handle non-photorealistic variations, we incorporate style-specific templates like *"art of..."* , *"an origami..."* , and *"...in a video game."*. This is particularly vital for benchmarks like ImageNet-R, ensuring that the textual prototype is not a rigid point estimate but encompasses a broader semantic volume covering artistic or abstract renditions.

### B.1.3. SCALE AND VIEWPOINT INVARIANCE

We utilize templates such as *"a large/small photo of..."* and *"itap of..."* (Internet slang for "I took a picture", implying a casual/realistic viewpoint). These descriptions help regularize the embedding against variations in object scale and camera angles, promoting spatially invariant recognition.

### B.2. VILA Algorithm

The algorithmic workflow of VILA consists of two primary phases: the **learning phase** (Algorithm 1), which constructs the model parameters, and the **inference phase** (Algorithm 2), which executes prediction.

As shown in Algorithm 1, the **learning phase** integrates gradient-based optimization with analytic recursive updates. *Lines 2-3* detail the specialized subspace construction. We initialize the specialized branch with a pre-trained ViT and adapters. The model is trained solely on the first task $\mathcal{D}_1$ via standard SGD. This is the only stage involving backpropagation. Once converged, the specialized backbone is permanently frozen. *Lines 4-10* outline the recursive analytic update. For the entire task stream $t = 1 \ldots T$ (including $\mathcal{D}_1$), VILA employs an incremental analytic learning protocol. For each batch, it extracts heterogeneous features from both branches and performs Unified Geometric Calibration (UGC). The analytic classifier weights $W$ are then updated recursively using the closed-form RLS solution.

As shown in Algorithm 2, the **inference phase** executes coarse-to-fine prediction. During inference, the model weights

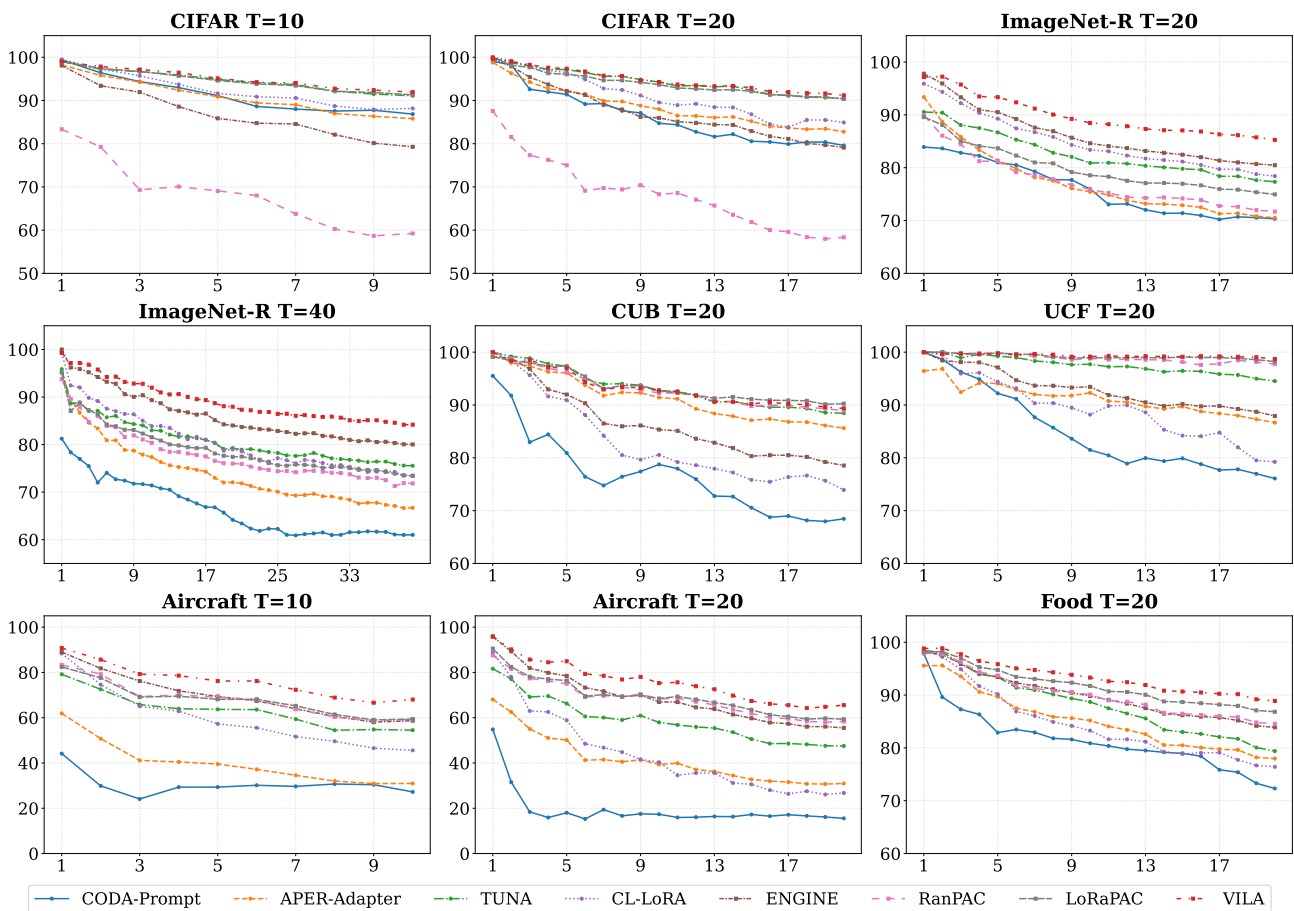

*Figure 8.* Detailed incremental accuracy curves across different benchmarks. The x-axis represents the number of tasks, and the y-axis denotes the last task accuracy $A_T$.

are fixed. The prediction process follows a dual-step refinement logic. *Lines 1-3* is analytic prediction. The test image is processed to obtain the calibrated feature $\mathbf{F}$. The analytic classifier outputs the primary logits $\hat{y}_{AL}$. *Lines 4-8* is semantic rectification. To mitigate the rigidity of the fixed backbone, the Candidate Semantic Enhancement (CSE) module selects the Top-$K$ probable classes and computes semantic similarity scores using CLIP text priors. The final prediction is the fusion of the analytic posterior and the semantic prior.

### B.2.1. PRACTICAL DEPLOYMENT AND OFFLINE SEMANTIC CACHING

While Algorithm 2 provides the theoretical formulation of the inference phase, we introduce an offline semantic caching strategy (VILA-OSC) to constrain the active parameter count and FLOPs during practical deployment. We explicitly decouple this optimization into two sub-processes:

*Offline semantic caching* (Algorithm 3). During the learning phase of any new task, the system extracts the text embeddings exactly once per novel class and caches them in a lightweight look-up table. Consequently, the heavy text encoder can be completely offloaded from the active computational graph.

*Highly efficient online prediction* (Algorithm 4). During continuous evaluation, the system simply retrieves these pre-computed prototypes in $O(1)$ time. This decoupling achieves high-fidelity multimodal rectification while reducing the text-encoding FLOPs to zero during actual deployment.

Hereafter, unless explicitly stated otherwise, all subsequent mentions and evaluations of VILA refer to this practical deployment implementation (VILA-OSC).

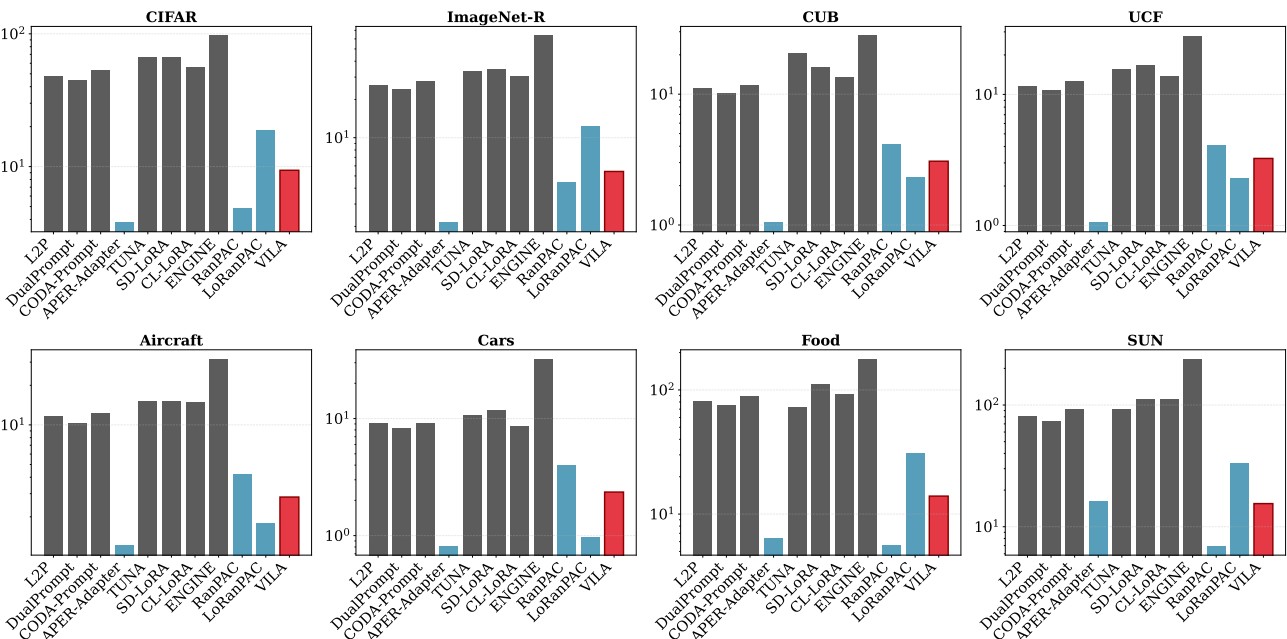

*Figure 9.* Detailed training time comparison. We report the total training time (in minutes) for all methods across 8 datasets ($T = 20$ for all). The y-axis is plotted on a logarithmic scale. VILA (red) consistently achieves low training latency, significantly outperforming traditional prompt learning methods (grey) and approaching the speed of the fastest adapters (blue).

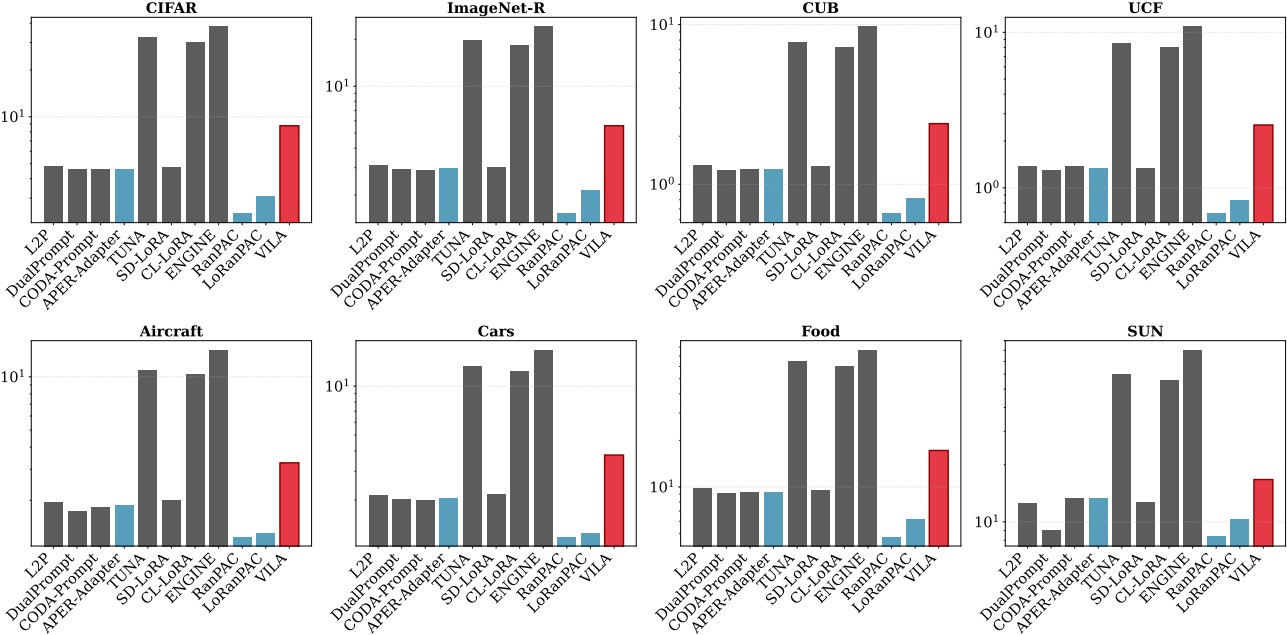

*Figure 10.* Detailed inference time comparison. We report the total inference time (in minutes) for all methods across 8 datasets ($T = 20$ for all). The y-axis is plotted on a logarithmic scale. VILA (red) maintains competitive inference speeds across all diverse datasets, avoiding the high computational costs associated with heavy baselines like CL-LoRA and ENGINE.

## C. Extended Benchmark Performance and Efficiency

### C.1. More Incremental Performance

To provide a granular view of the learning dynamics, we present the complete incremental accuracy curves across all evaluated settings in Figure 8. These experiments cover diverse scenarios, ranging from standard benchmarks (*e.g.*,

*Table 8.* Performance comparison (last-task accuracy $A_T$, average accuracy $\bar{A}$) under a matched computational budget. VILA demonstrates superior accuracy across 8 diverse benchmarks compared to RanPAC and LoRanPAC, even when utilizing equivalent visual backbones and adapters.

| Method | FLOPs Params | CIFAR $T = 20$ | ImageNet-R $T = 40$ | CUB $T = 20$ | UCF $T = 20$ | Aircraft $T = 20$ | Cars $T = 20$ | Food $T = 20$ | SUN $T = 50$ |
|---|---|---|---|---|---|---|---|---|---|
| RanPAC (ViT/Large) | 60.31G 306.47M | 91.11 94.55 | 80.43 85.25 | 88.21 93.16 | 97.12 98.78 | 54.40 64.27 | 71.12 73.42 | 86.77 91.55 | 80.97 87.09 |
| LoRanPAC (ViT/Large) | 60.31G 306.27M | 91.15 94.71 | 78.62 83.78 | 89.53 **93.55** | 98.40 99.21 | 56.56 66.35 | 72.84 81.84 | 88.31 92.51 | 81.91 87.85 |
| RanPAC (ViT-Adapter+ViT) | 33.96G 172.48M | 88.22 92.61 | 75.37 81.09 | 89.02 92.97 | 95.45 98.48 | 57.85 65.12 | 67.99 76.84 | 85.45 90.84 | 79.81 86.11 |
| LoRanPAC (ViT-Adapter+ViT) | 33.96G 172.48M | 84.81 89.33 | 73.17 78.55 | **90.29** 93.47 | 98.30 99.22 | 59.89 68.92 | 70.61 79.82 | 84.39 89.05 | 80.57 86.85 |
| VILA (ViT-Adapter+CLIP) | 33.98G 174.27M | **91.18** **94.72** | **84.20** **89.16** | 89.36 93.16 | **98.71** **99.35** | **65.56** **75.74** | **90.28** **94.19** | **88.89** **93.32** | **83.55** **89.53** |

CIFAR-100, ImageNet-R) to challenging fine-grained tasks (*e.g.*, CUB-200, FGVC-Aircraft) and varying sequence lengths ($T = 10, 20, 40$).

As observed, our proposed VILA consistently maintains superior accuracy throughout the incremental phases compared to SOTA baselines. Notably, in difficult fine-grained scenarios such as Aircraft $T = 20$ and long-sequence settings like ImageNet-R $T = 40$, where many competitive methods (*e.g.*, RanPAC, CODA-Prompt) exhibit sharp performance degradation, VILA demonstrates remarkable stability and effectively mitigates catastrophic forgetting.

## C.2. Full Efficiency Results

In this section, we provide the details of the computational efficiency for all compared methods across each of the 8 benchmark datasets. While the main text focuses on the average efficiency trade-off, the variability in dataset size (*e.g.*, ImageNet-R vs. CIFAR) and complexity (*e.g.*, number of classes) can affect methods differently.

*Training efficiency.* Figure 9 illustrates the total training time (in minutes) for each method on individual datasets. Consistent with the average results, VILA (red bars) demonstrates stable and efficient training performance across all scenarios. Notably, on large-scale datasets like Food and SUN, where heavy optimization-based methods (*e.g.*, SD-LoRA, ENGINE) suffer from explosion in training time (exceeding 60 minutes), VILA remains highly efficient ($\sim$15 minutes), comparable to lightweight adapter-based approaches. This confirms that VILA's training efficiency is robust to dataset scale and domain shifts.

*Inference efficiency.* Figure 10 presents the total inference time (in minutes) for each dataset. Inference latency is critical for real-world deployment. VILA maintains a low inference overhead, significantly faster than methods involving heavy ensemble or complex attention mechanisms (*e.g.*, CL-LoRA). Although simple linear probes or ultra-lightweight adapters (*e.g.*, RanPAC) achieve the lowest absolute inference time, VILA strikes a better balance by providing superior accuracy (as discussed in the main text) with only a marginal increase in latency.

## C.3. Computational Comparison under Matched Budget

As noted in the main text, baselines such as RanPAC and LoRanPAC inherently operate with smaller parameter and FLOP footprints in their default setups. To rigorously address this disparity and ensure that our performance improvements are not simply a byproduct of model capacity, Table 8 presents a comprehensive performance evaluation under strictly controlled computational constraints across all 8 diverse benchmarks.

First, to establish a fair baseline, we compare VILA against RanPAC and LoRanPAC utilizing equivalent visual backbones and adapters (ViT-Adapter+ViT, denoted as A+V). Even under this strictly matched 33.98G FLOPs footprint, VILA demonstrates overwhelming superiority, outperforming its counterparts in 7 out of 8 diverse benchmarks while remaining highly competitive on CUB. For instance, it exceeds LoRanPAC (A+V) by **+5.67%** in last-task accuracy ($A_T$) on the fine-grained Aircraft $T = 20$ benchmark.

Furthermore, to prove our gains are not merely capacity-driven, we deliberately grant the baselines a significantly larger

*Table 9.* Ablation study across the remaining 6 benchmarks. Metrics are last-task accuracy $A_T$ and average accuracy $\bar{A}$.

| Components | CIFAR ($T=10$) | ImageNet-R ($T=20$) | CUB ($T=10$) | UCF ($T=10$) | Food ($T=10$) | SUN ($T=30$) |
|---|---|---|---|---|---|---|
| Baseline | 90.79 / 94.22 | 76.72 / 82.15 | 88.72 / 93.10 | 97.50 / 98.73 | 86.74 / 91.36 | 78.55 / 85.52 |
| + CSE | 90.95 / 94.33 | 78.43 / 83.73 | 89.02 / 93.21 | 97.54 / 98.77 | 87.16 / 91.65 | 79.13 / 85.92 |
| + DB | 91.74 / 94.82 | 82.57 / 87.61 | 89.78 / 93.23 | 98.37 / 99.36 | 88.13 / 92.32 | 81.35 / 87.72 |
| + DB + UGC | 91.87 / 95.01 | 84.95 / 89.50 | 89.86 / 93.56 | 98.90 / 99.43 | 89.01 / 92.97 | 83.60 / 89.32 |
| VILA (Full) | **91.94 / 95.09** | **85.28 / 89.81** | **90.08 / 93.64** | **98.94 / 99.45** | **89.30 / 93.04** | **83.73 / 89.43** |

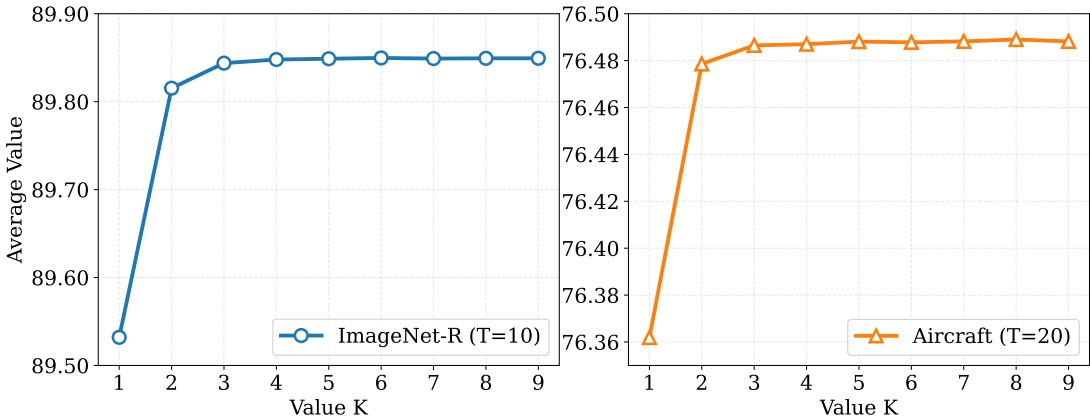

*Figure 11.* Ablation study (average accuracy $\bar{A}$) of the number $K$.

computational budget by equipping them with a massive ViT/Large backbone (306.47M parameters, far exceeding VILA). Remarkably, the lightweight VILA still surpasses this heavily-scaled LoRanPAC (ViT/Large) by **+5.58%** on ImageNet-R $T = 40$. This empirical evidence conclusively confirms that VILA's performance leap is strictly rooted in its structural innovation (specifically the decoupled dual-branch calibration), rather than brute-force scaling of parameters or FLOPs.

## D. More Ablation and Robustness Analysis

### D.1. Ablation Study on Remaining Benchmarks

While Table 5 in the main text provides ablation results for two representative fine-grained datasets (Aircraft and Cars), we here present the comprehensive ablation analysis for the remaining 6 benchmarks to further validate the contribution of each component across diverse data distributions. As shown in Table 9, we evaluate the progressive performance gains brought by Dual-Branch (DB), Unified Geometric Calibration (UGC), and Candidate Semantic Enhancement (CSE). The results consistently align with the observations in the main text.

*Generalization of DB.* The Dual-Branch architecture consistently provides the most significant boost, especially on complex datasets like ImageNet-R $T = 20$ and SUN $T = 30$. This underscores that anchoring a specialized branch with a universal vision-language anchor is a globally effective strategy for mitigating forgetting.

*UGC across various manifolds.* The Unified Geometric Calibration (UGC) is crucial for aligning the heterogeneous feature distributions. For example, on ImageNet-R, UGC alone improves $A_T$ from $82.57\%$ to $84.95\%$, proving that geometric alignment is essential regardless of the category semantics.

*Complementary nature of CSE.* Across all 6 additional benchmarks, the combination of DB, UGC, and CSE consistently achieves the peak performance. This confirms that the text-driven priors from the CLIP text encoder provide a stable "semantic rectification" that complements the calibrated visual features from the dual-branch backbone.

### D.2. Sensitivity to Hyperparameter $K$

In the Candidate Semantic Enhancement (CSE) module, the hyperparameter $K$ determines the size of the top-K candidate set selected from the primary analytic logits. Figure 11 visualizes the performance variations across different $K$ values.

We observe that the performance improves initially as $K$ increases from 1, and saturates rapidly at $K = 5$. Beyond this point, the performance exhibits negligible fluctuations ($< 0.1\%$). This empirical evidence demonstrates that VILA effectively extracts sufficient contextual information from a highly localized semantic neighborhood, and the overall framework remains remarkably robust to hyperparameter variations.

