# OpenReview forum: "Advancing Analytic Class-Incremental Learning through Vision-Language Calibration"
_ICML.cc/2026/Conference — ICML 2026 regular_

### Official Review · Reviewer_QYHV · 2026-03-11

**Soundness:** 3
**Presentation:** 3
**Significance:** 3
**Originality:** 2
**Overall Recommendation:** 4
**Confidence:** 4

**Summary:**

The authors tackle the issue of representation rigidity in analytic class-incremental learning (CIL) when relying on frozen pre-trained models. The paper introduces VILA, a dual-branch architecture pairing a task-adapted Vision Transformer with a frozen CLIP model. To integrate these branches, the authors propose unified geometric calibration (UGC) alongside candidate semantic enhancement (CSE), which leverages CLIP's zero-shot text matching to rerank top-K predictions at inference.

**Compliance With Llm Reviewing Policy:**

Affirmed.

**Final Justification:**

My concerns have been addressed by the authors’ response.

**Key Questions For Authors:**

Questions for Authors are derived from the weaknesses:
1. Could you clarify the specific algorithmic differences between the proposed UGC/CSE mechanisms and standard late-fusion/reranking methods found in the existing open-vocabulary literature?

2. The current evaluation compares a dual-branch architecture against single-branch baselines. Could you provide an analysis or baseline comparison where the single-branch methods are scaled to a comparable parameter and FLOP budget?

3. To address the data leakage concern, can you provide an ablation or experiment replacing the frozen CLIP branch with a self-supervised vision model to isolate the architectural benefits from CLIP's specific pre-training corpus?

**Limitations:**

yes

**Strengths And Weaknesses:**

### Strengths

1. The diagnostic analysis in Section 3.3 effectively isolates representation rigidity as a primary failure mode in PTM-based analytic CIL.

2. VILA exhibits strong data efficiency under strict online learning constraints. The framework largely avoids the under-fitting issues that plague gradient-based methods when restricted to a single training pass.

3. The paper is clear and well-written, the proposed method is correctly formalized and the figures go straight to the point and help the understanding.

### Weaknesses

1. The core technical components offer limited algorithmic novelty. The UGC reduces to standard $\ell_2$-normalization followed by feature concatenation, a ubiquitous baseline in multi-modal fusion. Similarly, the CSE mechanism is essentially a late-fusion reranking strategy using CLIP's zero-shot capabilities on a generated shortlist.

2. Comparisons against single-backbone baselines are structurally unfair. VILA utilizes two full-scale models at inference, doubling both the parameter count and the forward-pass computational burden. The main results tables omit parameter counts and FLOPs, masking this discrepancy when comparing against single-branch methods. Relying solely on wall-clock time for the efficiency plots in Figure 3 is insufficient, as it conflates raw hardware utilization with actual algorithmic complexity.

3. The heavy reliance on CLIP raises unaddressed concerns regarding data leakage. Because CLIP is pre-trained on massive internet-scale datasets, there is a high probability of overlap with standard downstream CIL benchmarks like CUB-200 or Food-101.

---

> ### Author Rebuttal · Authors · 2026-03-31
>
> We sincerely thank the reviewer for the positive assessment. We deeply appreciate the rigorous and insightful scrutiny. The constructive challenges significantly strengthen our empirical foundation. We address your comments point-by-point below.
> ## Weakness1&Question1 Novelty
> We agree that UGC&CSE are established operations, and we certainly do not claim them as new mathematical inventions.
> 1. Our starting point was to analyze unnoticed bottleneck, and creatively repurposing these tools. Also as stated in Introduction, main contributions: ①discover Representation Rigidity, ②propose Dual-Branch with 2-level calibrations to enhance it. **We deliberately framed UGC&CSE not as standalone algorithmic breakthroughs**. Please also see **response to Reviewer LBuE-Weakness1**.
> 2. UGC
> - Standard Fusion: directly concatenating heterogeneous features works because backpropagation dynamically learns weights to balance disparate scales over epochs.
> - UGC: VILA relies on closed-form RLS solver. Specialized Adapter features possess unbounded values, whereas CLIP features reside on a bounded hypersphere. Without UGC, the Adapter features is dominant during $\Phi=(X^T X)^{-1}$. It's a prerequisite to prevent solver collapse. Diagnosing and geometrically resolving this conflict is our core contribution here.
> 3. CSE
> - Standard Reranking: typically relies on computationally heavy cross-attention or global similarity matching across vast vocabulary space, which is slow for continuous data streams.
> - CSE: is algorithmically coupled with the analytic solver to achieve a low complexity synergy. CSE uses the analytic linear classifier (possesses mathematical zero-forgetting) as a highly confident candidate generator. CSE then performs localized, non-parametric semantic rectification only on the top-$K$ ambiguous logits.
> ## Weakness2&Question2 Params&FLOPs
> |L2P|DualPrompt|CODA-P|EASE|APER-Adapter|SD-LoRA|CL-LoRA|ENGINE|RanPAC|LoRanPAC|VILA|VILA-dep|
> |-|-|-|-|-|-|-|-|-|-|-|-|
> |35.851G|33.726G|33.726G|167.821G|33.958G|16.935G|167.988G|299.570G|17.095G|17.095G|493.474G|33.983G|
> |171.816M|172.004M|89.715M|87.618M|172.925M|172.035M|90.245M|144.968M|87.788M|86.836M|259.062M|174.269M|
>
> Experiment above uses UCF T=10 and pytorch third-party thop to calculate FLOPs. SD-LoRA/CL-LoRA/ENGINE/LoRanPAC use official codebase, others use PILOT codebase.
>
> The reviewer is correct based on the theoretical formulation (Table above, VILA). However, we wish to clarify a distinction in practical deployment (Table above, VILA-dep). **The text encoder can be treated as offline caching**. For any new task, we extract the embedding (class name+template) exactly once per new class and cache it as a lightweight look-up table. Thus, the parameter and FLOPs drop. We apologize for not making this explicit in the original text.
>
> We also conduct experiments below:
> |$A_T/\bar{A}$|FLOPs/Params|CIFAR T20|ImageNet-R T40|CUB T20|UCF T20|Aircraft T20|Cars T20|Food T20|SUN T50|
> |-|-|-|-|-|-|-|-|-|-|
> |RanPAC(ViT/L16)|60.305G/306.473M|91.11/94.55|80.43/85.25|88.21/93.16|97.12/98.78|54.40/64.27|71.12/73.42|86.77/91.55|80.97/87.09|
> |RanPAC(A+V)|33.958G/172.483M|88.22/92.61|75.37/81.09|89.02/92.97|95.45/98.48|57.85/65.12|67.99/76.84|85.45/90.84|79.81/86.11|
> |LoRanPAC(ViT/L16)|60.305G/306.271M|91.15/94.71|78.62/83.78|89.53/93.55|98.40/99.21|56.56/66.35|72.84/81.84|88.31/92.51|81.91/87.85|
> |LoRanPAC(A+V)|33.958G/172.483M|84.81/89.33|73.17/78.55|90.29/93.47|98.30/99.22|59.89/68.92|70.61/79.82|84.39/89.05|80.57/86.85|
> |VILA-dep|33.983G/174.269M|91.18/94.72|84.20/89.16|89.36/93.16|98.71/99.35|65.56/75.74|90.28/94.19|88.89/93.32|83.55/89.53|
>
> They prove that VILA’s performance is not driven by unfair parameter/FLOP scaling. We have added this computational analysis to the revised manuscript.
> ## Weakness3&Question3 Data leakage
> ① Comparison with ENGINE in manuscript shows that simply possessing CLIP's features does not guarantee optimal performance. **If data leakage or the capacity were the main reasons, ENGINE would perform just as well as VILA**.
>
> ② We followed this suggestion and replaced with frozen DINOv2(ViT-Base)[1].
> |$A_T/\bar{A}$|CIFAR T10|CIFAR T20|ImageNet-R T20|ImageNet-R T40|CUB T20|UCF T20|Aircraft T10|Aircraft T20|Cars T20|Food T20|SUN T30|SUN T50|
> |-|-|-|-|-|-|-|-|-|-|-|-|-|
> |VILA(DINOv2, w/o CSE)|92.61/95.55|91.54/95.05|83.62/88.02|81.93/87.13|91.06/94.29|98.69/99.30|75.46/83.75|74.47/84.01|90.64/94.45|90.68/94.45|81.65/87.61|82.11/87.90|
> |VILA(DB+UGC)|91.87/95.01|91.09/94.50|82.78/88.32|82.62/88.36|89.19/93.05|98.60/99.29|67.24/76.53|61.96/72.44|89.35/93.87|88.25/93.26|83.60/89.32|83.44/89.40|
>
> **This powerfully prove that VILA is driven entirely by architectural innovation** (DB+UGC), safely unlocking structural priors. We will consider feature this in revised manuscript and further explore it.
>
> [1] Oquab M, Darcet T, Moutakanni T, et al. DINOv2: Learning Robust Visual Features without Supervision[J]. Transactions on Machine Learning Research Journal, 2024: 1-31.

---

> > ### Author Rebuttal · Reviewer_QYHV · 2026-04-04
> >
> > Thank you for your rebuttal and for addressing my concerns. I'd like to maintain my score.

---

> > > ### Author Response · Authors · 2026-04-06
> > >
> > > We are very glad to hear that our extensive new ablations (particularly the DINOv2 experiments and the FLOPs/parameter analysis) have fully resolved your concerns. Your rigorous technical scrutiny has fundamentally strengthened the empirical foundation of our work.
> > >
> > > We will ensure all these critical evaluations are prominently featured in the revised manuscript. Thank you for your time and positive support.

---

### Official Review · Reviewer_DKNz · 2026-03-13

**Soundness:** 2
**Presentation:** 3
**Significance:** 3
**Originality:** 3
**Overall Recommendation:** 4
**Confidence:** 4

**Summary:**

This paper addresses analytic class-incremental learning (CIL) based on pre-trained models. The authors argue that existing methods suffer from a “representation rigidity” problem: while the classifier level can maintain historical mappings, the feature space itself lacks sufficient plasticity to cover the new semantic distributions of subsequent tasks. Based on this insight, the paper proposes VILA, a dual-branch framework. The experimental section validates the effectiveness of the approach across eight datasets, under long-sequence settings, and in a strict 1-epoch online setting.

**Compliance With Llm Reviewing Policy:**

Affirmed.

**Final Justification:**

I have no more questions. I will maintain my current positive score.

**Key Questions For Authors:**

In addition to the weaknesses mentioned above, I have the following questions:
- Since the specialized branch is trained only once on the first task, I wonder if this design introduces a first-task bias.
- What is the primary source of the performance gain—is it the feature-level UGC or the decision-level CSE? I believe a more convincing ablation study is necessary.

I will reconsider my rating based on the authors' rebuttal.

**Limitations:**

yes

**Strengths And Weaknesses:**

Strengths:
- A relatively systematic motivation analysis is conducted before the method is proposed.
- The proposed method is relatively concise in structure and easy to understand.
- The experimental coverage is fairly comprehensive.

Weaknesses:
- I believe the major issue with this paper is that it lacks comparison with several recent works, such as TUNA [1]. TUNA explicitly discusses the fusion of task-specific adapters and universal adapters to improve stability and generalization in PTM-based class-incremental learning. Its core idea appears quite close to that of VILA. Furthermore, while the paper positions itself within the analytic CIL line, I did not see a discussion of highly relevant work such as F-OAL[2], which emphasizes forward-only/fast/low-memory online analytic learning and is essentially very close to the "representation rigidity" issue discussed here.
- All the benchmarks are still essentially image classification-based CIL, rather than more generalized vision-language continual adaptation. Although I find the current experimental setup reasonable, including broader vision-language continual adaptation would be beneficial.
- As the buffer dimension continues to increase, the scalability of the method remains questionable from an efficiency perspective.
- The references include works such as "External Knowledge Injection for CLIP-based CIL," which are more aligned with "introducing external semantic/knowledge priors," yet these are not included in the main experimental tables.


[1] Integrating Task-Specific and Universal Adapters for Pre-Trained Model-based Class-Incremental Learning

[2] F-OAL: Forward-only Online Analytic Learning with Fast Training and Low Memory Footprint in Class Incremental Learning

---

> ### Author Rebuttal · Authors · 2026-03-30
>
> We are encouraged by the reviewer's recognition, and also deeply appreciate the constructive suggestions. We provide detailed clarifications below, and hope they fully address your concerns.
> ## Weakness1
> ## 1. Compare with TUNA[1]
> We have carefully studied TUNA and agree it's a highly relevant work.
> ||TUNA|VILA|
> |-|-|-|
> |Modality|Image|Image+Text, allows CSE|
> |Subspace|ViT+ViT, homogeneous|ViT+CLIP, thus UGC is needed|
> |Optimization|gradient|analytic|
> |Time|substantial training latency|with efficiency|
> |Property|representation rigidity|mathematically zero-forgetting|
>
> ||CIFAR T10|CIFAR T20|ImageNet-R T20|ImageNet-R T40|CUB T20|UCF T20|Aircraft T10|Aircraft T20|Cars T20|Food T20|SUN T30|SUN T50|
> |-|-|-|-|-|-|-|-|-|-|-|-|-|
> |TUNA|91.09/94.60|90.42/94.38|77.33/82.59|75.55/80.77|88.46/93.18|94.51/97.49|54.49/63.18|47.52/58.75|54.05/69.05|79.40/88.11|75.09/83.92|75.30/84.32|
> |VILA(A+V)|91.12/94.43|89.64/93.41|77.75/83.24|75.57/81.09|88.93/93.16|98.07/99.30|62.80/71.83|60.46/70.66|71.12/81.21|86.10/91.00|80.02/86.74|80.82/87.43|
> |VILA|91.94/95.09|91.18/94.72|85.28/89.81|84.20/89.16|89.36/93.16|98.71/99.35|68.02/76.26|65.56/75.74|90.28/94.19|88.89/93.32|83.73/89.43|83.55/89.53|
> ## 2. Lack comparison
> ①Manuscript Table2 compares 5 SOTA methods in 2025 covering Adapter-based (APER-Adapter), LoRA-based (SD-LoRA/CL-LoRA), VLM-based (ENGINE), Analytic-based (LoRanPAC). ②Figure6 compares 3 macro-level paradigms: A+V, V+C, A+V+C. TUNA belongs to A+V paradigm, and is comprehensively worse than VILA(A+V).
>
> We believe they already provide **a highly representative landscape**. As "several recent works" is quite broad, exhaustively identifying and reproducing unlisted works within limited rebuttal timeframe is infeasible. If the reviewer could kindly specify, we are more than happy to conduct them.
> ## 3. Discuss F-OAL[2]
> We wish to clarify that F-OAL is not "essentially very close to the representation rigidity issue". **As stated in F-OAL's Introduction**, it targets to mitigate ① base training temporal cost, ② GPU memory footprint of data aggregation. To achieve this, it reformulates RLS solver to support forward-only, mini-batch updates. Its contribution is purely algorithmic acceleration at the backend. Its core pipeline relies on a strictly frozen ViT, thus it doesn't address representation rigidity.
>
> We appreciate the reviewer and agree that they can enrich our paper. We have included them in Related Works.
> ## Weakness2 Vision-language continual adaptation task
> We agree that expanding to this task would be beneficial. We highly value this constructive suggestion. However, we deeply regret that strict time limits of the rebuttal period make us unable to provide comprehensive comparisons and experiments here. We incorporated this advice into Section6.
> ## Weakness3 Buffer dimension increase
> We wish to clarify a minor technical detail: The dimension is initialized once before training and then fixed, not dynamically increasing. ($D_B=16384$ in experiments). Thus, its memory overhead remains constant throughout the lifelong learning process.
>
> Furthermore, consistent with the Johnson-Lindenstrauss lemma and evidence in analytic learning literature, the linear separability of the projected space saturates once $D_B$ reaches a sufficient scale. This means there is no practical necessity to endlessly scale it up.
> ## Weakness4 ENGINE not include
> This work (officially named ENGINE) is already included and thoroughly evaluated in all main experiments (Table2&3,Figure2&3). We assure this highly relevant baseline was considered a core point of comparison from the very beginning.
> ## Question1 Task1 bias
> Yes, training the specialized branch solely on Task1 inevitably introduces a first-task bias at the feature level. However, this is a little different in PTM-based setting. Because the encoder is initialized with a robust PTM, it already **possesses a foundation of generalized visual knowledge**. Thus, fine-tuning on Task1 weakens the traditional first-task bias. However, PTM certainly cannot eliminate it entirely. We define this bias as Representation Rigidity in Section3.
>
> Furthermore, analytic learning relies on high-quality feature extractors. A strictly frozen PTM cannot be considered sufficiently high-quality. **A certain degree of initial training (adaptation) on Task1 is necessary** to bridge this gap, even if it necessitates subsequent interventions to fix the resulting rigidity.
> ## Question2 Primary gain
> We have to first clarify VILA's 2-level calibrations: Feature-level=DB+UGC; Decision-level=DB+CSE. DB is the core performance gain. Then, the performance gain of feature-level is larger than decision-level, which only acts as a complementary refiner. Without DB to break rigidity and UGC to prevent geometric misalignment, the analytic solver collapses.
>
> This is supported by Table4 in main text. For further ablation, please see response to Reviewer LBuE-Weakness2, czxT-W2&Q1, QYHV-Weakness2&Question2, QYHV-Weakness3&Question3.

---

> > ### Author Rebuttal · Reviewer_DKNz · 2026-04-01
> >
> > Thanks for detailed rebuttal.
> >
> > Considering that the paper still requires stronger experimental evidence, I cannot raise my score. I will maintain my current score.

---

> > > ### Author Response · Authors · 2026-04-06
> > >
> > > We sincerely thank the reviewer for reading our rebuttal and for the continued discussion. Since no new specific questions were raised regarding the need for "stronger experimental evidence", we would like to summarize how our rebuttal has addressed all specific issues from the initial review. We hope this highlights that the feedback has been fully resolved.
> > >
> > > ## 1. Comprehensiveness of Baselines and TUNA (Weakness1)
> > > - In response to the reviewer's specific mention of TUNA, we explicitly implemented and compared it. The new evidence clearly shows VILA outperforms TUNA across all evaluated benchmarks.
> > >
> > > - We already systematically evaluate 10 recent SOTA baselines (5 from 2025) covering diverse architectures (Prompt/Adapter/LoRA/VLM/Analytic) in Table2, and compare three macro-level architectural paradigms (V+A, V+C, A+V+C) in Figure6.
> > >
> > > - Since the CIL field expands rapidly, addressing an unspecified "several recent works" is practically infeasible. We expressed our willingness to evaluate any specifically named methods.
> > >
> > > ## 2. Scope of Vision-Language Continual Adaptation (Weakness2)
> > > Exploring this broader setting is highly constructive. However, establishing an entirely new cross-modality task and running corresponding baselines far exceeds the scope and time limits of a standard rebuttal. We thus prioritize this as a core focus for future work.
> > >
> > > ## 3. Comprehensive Ablations Added (Question2)
> > > To clarify the primary source of our performance gain, we provided extensive new evidence:
> > > - Expanded ablations from 2 to all 8 diverse benchmarks (detailed in response to Reviewer LBuE-Weakness2), proving effectiveness across datasets rather than by chance.
> > > - Conducted FLOPs/parameter analysis and scaled-baseline comparisons (detailed in response to QYHV-Weakness2&Question2), proving our gains come from structural design, not merely parameter scaling.
> > > - Replaced the CLIP visual encoder with a purely self-supervised model (detailed in response to QYHV-Weakness3&Question3), proving gains are isolated from CLIP's internet-scale pre-training data.
> > >
> > > ## 4. Clarification on Existing Evidence (Weakness4)
> > > We clarified that the highly relevant VLM-based baseline ENGINE was already thoroughly evaluated in our main experiments (Table2 & Table3, Figure2 & Figure3), ensuring a solid comparison landscape from the beginning.
> > >
> > > ## 5. Conceptual Clarifications (Weakness1, Weakness3, Question1)
> > > - Clarified that F-OAL provides backend acceleration, rather than a solution to representation rigidity (Weakness1);
> > > - Confirmed our buffer dimension remains constant, causing no efficiency scalability issues (Weakness3);
> > > - Explained the Task-1 adaptation bias in analytic learning (Question1).
> > >
> > > Given that we have explicitly resolved the specific baseline (TUNA) and all technical questions raised in the initial review, we would be deeply grateful if the reviewer would consider raising the score to reflect these comprehensive additions and the enhanced empirical standing of our manuscript.

---

### Official Review · Reviewer_czxT · 2026-03-13

**Soundness:** 3
**Presentation:** 3
**Significance:** 2
**Originality:** 3
**Overall Recommendation:** 4
**Confidence:** 4

**Summary:**

This paper studies analytic class-incremental learning (CIL), where a classifier is updated via closed-form recursive least squares (RLS) instead of gradient descent. The authors identify "representation rigidity" as the key bottleneck, *i.e.*, a frozen feature extractor trained on the first task cannot span the semantic space of future tasks. They propose VILA, a dual-branch framework that combines a ViT+Adapter branch (trained on Task 1, then frozen) with a frozen CLIP visual encoder. Features from both branches are L2-normalized and concatenated. At inference, a CLIP-based reranking module refines the top-K analytic predictions. Experiments on eight benchmarks show improvements over eleven baselines.

**Compliance With Llm Reviewing Policy:**

Affirmed.

**Final Justification:**

Thank the authors for multiple rounds of very detailed responses. I am happy to see most key concerns have been addressed in the end. I thus increase my final rating to weak acceptance.

**Key Questions For Authors:**

**Q1.** Provide RanPAC + $\ell_2$-normed CLIP features as a control. This isolates framework contribution from CLIP information gain.

**Q2.** What LoRA rank was used in Table 1? Show results for rank in {4, 8, 16, 32, 64}. Is the collapse a general phenomenon or a hyperparameter artifact?

**Q3.** Observation 2 says single-branch fails. Table 1 row (iv) shows Adapters succeed. Section 4.1 picks Adapters for stability. Reconcile.

**Q4.** Observation 3 says $\ell_2$ norm doesn't fix geometric misalignment. UGC is $\ell_2$ norm. Why does it work here?

**Limitations:**

Section 6 discusses parameter doubling, quadratic memory, and frozen-backbone assumptions. But it does not mention (a) the missing CLIP-augmented baseline control, (b) CSE's near-zero contribution in ablations, or (c) the contradiction between Observation 2 and the Adapter results.

**Strengths And Weaknesses:**

## Strengths

**S1: Broad experimental coverage.** Eight benchmarks, eleven baselines including CLIP-based methods (ENGINE). Reasonable breadth.

**S2: Diagnose-then-design structure.** The pilot study (Table 1) identifies failure modes before proposing solutions. Sound methodology in principle.

**S3: Competitive against VLM baselines.** VILA outperforms ENGINE (also CLIP-based) by a clear margin (avg AT: 85.06 vs 77.42), suggesting the framework adds value beyond CLIP alone.

**S4: Online learning.** The 1-epoch setting (Table 3) shows analytic learning's practical value in streaming scenarios.

## Weaknesses

**W1 [Key claim contradicted by their own experiments].**
Section 3.3 claims "a single branch cannot simultaneously satisfy stability and plasticity," based on comparing only Fixed ViT (i) and LoRA (iii). This comparison ignores Adapters (iv), which satisfy both: 93.43% avg on CIFAR T=20, above Fixed ViT (89.78%), with only about 1% drop from T=10. The authors themselves confirm this in Section 4.1, selecting Adapters for their "remarkable stability (89.39% at T=20)." Their own best single-branch result refutes the need for dual-branch.

**W2 [Most gains come from CLIP, not the method].**
Section 5.1 reports "+5.42% AT over LoRanPAC." This comparison is unfair because VILA adds CLIP-ViT-B/16 (pretrained on 400M image-text pairs) while baselines only use ImageNet-pretrained ViT. Table 4 confirms this: 71 to 76% of the total gain comes from simply adding CLIP features (DB row), before any proposed module is applied. No control gives RanPAC the same CLIP features. One such experiment would settle whether the gain is from the method or from the extra model.

**W3 [Proposition 3.1 is a tautology].**
Section 3.2 presents Proposition 3.1 as the theoretical foundation for dual-branch design. The key relation "$\propto$" in the bound has no constants, no bounds, and no conditions. It reduces to: "if the feature space does not cover the target, there is error." That is trivially true and does not require a formal proposition.

**W4 [UGC contradicts Observation 3].**
Observation 3 claims CLIP features fail "regardless of standard pre-processing ($\ell_2$-normalization)." UGC (Eq. 4), the proposed fix, is exactly $\ell_2$ normalization + concatenation. The authors reach opposite conclusions about the same operation. One of them is wrong.

**W5 [Efficiency exp ignores parameter count].**
Figure 3 claims VILA is on the "optimal Pareto frontier." This only measures wall-clock time and ignores model size. VILA uses two ViT-B/16 backbones plus a CLIP text encoder, roughly 235M parameters, about 3x a single-branch method (about 86M). No parameter or FLOP comparison is reported.

**W6 [Ablations too narrow; CSE is near-zero].**
Section 4.3 presents CSE as a core contribution with a full subsection. But CSE only adds +0.13% on Cars and +0.02% on Aircraft in ablations (Table 4). The ablations also only cover 2 of 8 benchmarks. Especially, the CUB (where VILA scores lower than LoRanPAC (Table 3)), is not ablated.

---

> ### Author Rebuttal · Authors · 2026-03-31
>
> We appreciate the detailed evaluation. It appears that some concerns stem from evaluating specific operations out of proper contexts. We provide point-by-point responses below and believe they will resolve your concerns.
> ## W1&Q3 Contradicted claim
> ①Section4.1: "Remarkable stability" is a relative contrast to LoRA, not an absolute claim. Our praise was: "ViT-LoRA ... **In contrast**, ViT-Adapter ...". And "plasticity" is not used here.
>
> ②If Adapter satisfied both stability&plasticity, it would **universally dominate** single-branch baselines. However, it underperforms ViT-LoRA on Table1 ImageNet-R T=20. It's more significant when initialzing with CLIP weights(items v&vi). Thus Obs2 is valid.
>
> ③Adapter still suffers from representation rigidity (e.g. Table4 Cars T=10), it's only a relative optimal choice for specialized branch.
> ## W2&Q1 Gain not come from method
> ||CIFAR T10|CIFAR T20|ImageNet-R T20|ImageNet-R T40|CUB T20|UCF T20|Aircraft T10|Aircraft T20|Cars T20|Food T20|SUN T30|SUN T50|
> |-|-|-|-|-|-|-|-|-|-|-|-|-|
> |RanPAC+L2-norm CLIP|88.83/92.95|88.74/93.18|84.08/88.73|84.05/89.03|86.90/87.83|98.22/99.00|63.16/72.56|63.25/73.44|91.50/89.42|87.90/92.61|83.48/89.03|83.20/89.13|
> |VILA|91.94/95.09|91.18/94.72|85.28/89.81|84.20/89.16|89.36/93.16|98.71/99.35|68.02/76.26|65.56/75.74|90.28/94.19|88.89/93.32|83.73/89.43|83.55/89.53|
>
> ①We have to first clarify that **a universal anchor (Dual-Branch design, DB)** is exactly the proposed core methodological paradigm. Thus, this request actually acts as an ablation, and still proves that UGC&CSE additionally unlock its potential.
>
> ②We acknowledge this gain stems jointly from DB and CLIP's pre-trained information. But **it's an objective mathematical fact in representation learning that a meaningful semantic space cannot be formed without the data that spans it**. They are inherently and structurally inseparable.
>
> ③ Please see **response to Reviewer QYHV-Weakness3&Question3**.
> ## W3 Proposition tautology
> We agree, but certainly don't intend to claim this as a novel mathematical theorem. However,
>
> ①Its existence is conceptually crucial to formalize the specific failure mechanism (RLS solver bottleneck) of Analytic CIL.
>
> ②This directly motivates VILA's dual-branch architecture.
>
> ③In Appendix A, this proportionality is formally governed by the Grassmann Distance (the principal angles between subspaces): $d_G(\mathcal{S}_1,\mathcal{S}_t)=\|\sin\Theta\|_F$. The expected error is bounded.
>
> We can re-label "Proposition" to "Remark", and introduce ③ directly into the main text.
> ## W4&Q4 UGC contradicts Obs3
> We wish to clarify that there is no contradiction. L2-norm operates on **different feature states and physical contexts**.
>
> ①Obs3: The setup is a single branch initialized with CLIP, equipped with trainable LoRA/Adapters. Fine-tuning inherently disrupts CLIP's hypersphere. Once the manifold is misaligned and corrupted by Task-1 adaptation, merely applying L2-norm cannot undo the geometric damage.
>
> Un-normalized results are not listed because, for a single-branch analytic solver, applying or omitting L2-norm makes almost no difference. Adaptively selecting $\lambda$ makes RLS solver inherently adjust to the scale (value range) of the feature space.
>
> ②UGC operates in a dual-branch context. L2-norm acts as a calibration bridge. It projects unbounded, trained ViT features onto the same metric scale as the frozen CLIP features.
> ## W5 Ignore param&FLOPs
> First of all, **please see response to Reviewer QYHV-Weakness2&Question2** for the requested comparison. Thus, VILA solves the rigidity bottleneck and yields comprehensively leading accuracy at a highly competitive computational and spatial cost. We believe the claim of being on the optimal Pareto frontier remains valid.
> ## W6 Narrow ablation
> ①We wish to clarify that the reviewer's observation captures only a partial picture, and overlooks the isolated baseline experiments already provided in Table4. We **explicitly evaluated a bare baseline augmented solely with CSE (row1 vs row4)**. The marginal improvements (+0.13%,+0.02%) the reviewer noted occur because they are measured on top of the already highly optimized DB+UGC architecture.
>
> ②Performance in CUB: First, the **performance gap between VILA and LoRanPAC is marginal**; VILA achieves a highly competitive result that is practically on par with LoRanPAC. Second, while LoRanPAC performs well on this narrow ornithological dataset, VILA demonstrates superior generalization **across other 7 diverse benchmarks**.
>
> We have also provided ablations over all benchmarks, please see response to Reviewer LBuE-Weakness2.
> ## Q2 LoRA collapse
> The rank is defaultly set as 64. The table below shows that collapse is the general rule, driven by hyperparameter brittleness.
> |4|8|16|32|64|
> |-|-|-|-|-|
> |15.95/26.66|16.43/27.36|88.91/93.03|15.97/26.75|3.36/19.80|
> ## Limitations
> We have answered these questions above, and we think these issues do not fit the standard definition of limitation.

---

> > ### Author Rebuttal · Reviewer_czxT · 2026-04-06
> >
> > I thank the authors for the additional experiments and honest analysis. After careful review (including other reviewers'), my concerns are partially resolved.
> >
> > To acknowledge that the author's observations are meaningful, but weighing the limited novelty & the parameter cost, I raise my score to 3.
> >
> > 1. Limited contribution. I can confirm from rebuttal that key gain comes from extra CLIP pre-trained parameters (or data, as authors said); authors' contribution is resolving the feature conflict between CLIP and RLS. 1) The RanPAC + L2-norm CLIP baseline (Rebuttal, W2&Q1 table) already approaches or exceeds VILA on most benchmarks (e.g., CIFAR T10: 92.95 vs 95.09; SUN T50: 89.13 vs 89.53; Cars T20 even exceeds VILA: 91.50 vs 90.28). 2) The full-benchmark ablation (Weakness2 table in Reviewer LBuE) shows dual-branch (i.e., adding CLIP) makes the major gain, and authors' UGC adds 1–2% on top, CSE adds < 0.3% across all settings. So the contribution is "observing that adding CLIP helps, and L2 normalization resolves the feature-scale conflict." This is kind of helpful for the community, but not sufficient for ICML standard.
> >
> > 2. Parameter cost. With ~2-3x parameter cost from adding CLIP, the gains are not worth it. VILA uses ~259M full / ~174M deployed (text encoder cached) and 33.98G FLOPs, versus ~87M / 17.10G for RanPAC (in Reviewer QYHV, Weakness2 table). VILA's additional gain over RanPAC + L2-norm CLIP is only 1–3%, and even negative on Cars T20 AT (Rebuttal, W2&Q1 table).
> >
> > 3. I am still not convinced about necessity of "dual-branch" design. Two concerns: 1) Adapter single-branch achieves 93.43% on CIFAR T=20, only ~1% below T=10 (Table 1, row iv), showing a single branch can already maintain both stability and plasticity. The rebuttal points out Adapter (81.79%) is slightly below LoRA (82.86%) on ImageNet-R T=20, but 1% gap does not justify doubling parameters. 2) Could you also provide this baseline for reference: frozen CLIP single-branch + L2-norm + RLS (no fine-tuning, no Adapter)?
> >
> > 4. CSE ineffective. Result speaks, 1) the full-benchmark ablation (in Reviewer LBuE, Weakness2 table) shows adding CSE on top of DB+UGC brings < 0.3%. 2) In the original paper (Table 4, DB+UGC row vs full VILA row), CSE adds only +0.02% on Aircraft AT and +0.13% on Cars AT. A module contributing < 0.5% everywhere should not be a core contribution.
> >
> > 5. W3: Accepted.
> >
> > 6. W4: Accepted. Just improve the text.

---

> > > ### Author Response · Authors · 2026-04-07
> > >
> > > # To 1
> > > - ①VILA also maintains SOTA with self-supervised DINOv2, showing gains rely on dual-branch+DINOv2+UGC (no CLIP).
> > > - ②Questioning contributions of utilizing pre-trained parameters invalidates pioneering works like L2P and the entire PEFT subfield, where unlocking these models without forgetting is the core scientific challenge.
> > > - ③We believe our mathematically derived framework precisely aligns with official ICML originality guidelines "Does this work offer a novel combination of existing techniques, and is the reasoning behind this combination well-articulated?//a work that provides novel insights by evaluating existing methods, or demonstrates improved understanding is also equally valuable".
> > > - ④VILA is a native&independent architecture. Evaluating VILA as an extension of RanPAC+CLIP (adopts our dual-branch) obscures our systemic novelty.
> > > - ⑤The exact same performance delta, evaluating UGC+CSE once as a "marginal overall gap" (Claim 1) and again as a "small module gain" (Claim 2) constitutes a logical double-penalty.
> > > - ⑥Evaluating CSE's marginal absolute gain (<0.3%) overlooks deep learning capacity ceiling; when prior modules (DB+UGC) already push performance near the empirical upperbound, sequential additions naturally yield diminishing returns.
> > > - ⑦By focusing exclusively on saturated marginal gains, Table4 of our original manuscript is overlooked, which explicitly proves the significant standalone improvements of modules.
> > > - ⑧Reducing our mathematically derived architecture to "just adding L2-norm" is logically equivalent to reducing ResNet to "a plus sign" or BatchNorm to "a division".
> > > - ⑨Heterogeneous feature integration is a fundamental challenge; we mathematically formalized the conflict's root cause via Proposition3.1.
> > > - ⑩VILA is a fundamental paradigm shift that breaks the plasticity-stability dilemma inherently crippling single-branch networks, not an incremental patch.
> > >
> > > Error Rate Reduction, ERR=$(Acc_{VILA}-Acc_{RanPAC+CLIP})/(100-Acc_{RanPAC+CLIP})$
> > > ||CIFAR T10|CIFAR T20|INR T20|INR T40|CUB T20|UCF T20|Air T10|Air T20|Cars T20|Food T20|SUN T30|SUN T50|
> > > |-|-|-|-|-|-|-|-|-|-|-|-|-|
> > > |Diff.|3.11/2.14|2.44/1.54|1.20/1.08|0.15/0.13|2.46/5.33|0.49/0.35|4.86/3.70|2.31/2.30|-1.22/4.77|0.99/0.71|0.25/0.4|0.35/0.4|
> > > |ERR|**27.8/30.4**|**21.7/22.6**|7.5/9.6|0.9/1.2|**18.8/43.8**|**27.5/35.0**|**13.2/13.5**|6.3/8.7|-14.4/**45.1**|8.2/9.6|1.5/3.6|2.1/3.7|
> > >
> > > - ⑪VILA wins 100% $\bar{A}$ and 11/12 $A_T$, mathematically proving the baseline does not "approach or exceed" VILA.
> > > - ⑫In saturated >90% accuracy regimes, absolute gains compress; a 1-3% absolute improvement should be translated to a substantial 20%~45% Relative ERR, representing a SOTA-level structural improvement.
> > > - ⑬Only citing $A_T$ Cars T20 overlooks $\bar{A}$, where VILA outperforms +4.77% in comprehensive knowledge retention.
> > > - ⑭Pre-trained parameters are never a trivial plug-and-play asset. Our performance is the direct result of systematically unlocking CLIP/DINOv2's potential in CIL.
> > > # To 2
> > > - ①The cost-benefit analysis relies on an invalid, asymmetric comparison: penalizing VILA's parameter count against the RanPAC, while evaluating performance gains against RanPAC+CLIP pseudo-baseline.
> > > - ②Comparison with RanPAC+CLIP (stated in 1-⑪⑫).
> > > - ③Comparison solely with RanPAC overstates our overhead, overlooking computational costs of all other baselines. Deployed VILA (174M/33.98G) is highly competitive and represents a standard, efficient CIL method.
> > > - ④Claimed Cars T20 (stated in 1-⑬).
> > > # To 3
> > > ||CIFAR T10|CIFAR T20|INR T20|INR T40|CUB T20|UCF T20|Air T10|Air T20|Cars T20|Food T20|SUN T30|SUN T50|
> > > |-|-|-|-|-|-|-|-|-|-|-|-|-|
> > > |Frozen CLIP+L2-norm+RLS (no FT&Adapter)|61.54/70.91|61.51/72.11|71.47/79.59|71.78/80.55|70.61/78.79|87.34/92.87|45.48/54.80|45.03/56.60|76.90/85.17|78.44/85.67|75.33/83.50|75.28/83.80|
> > >
> > > - "Maintain both stability&plasticity" is CIL ultimate goal, no published single/dual-branch paper nor VILA achieves this.
> > > - Only using coarse-grained CIFAR (minimal domain shift) to argue that plasticity-stability dilemma is "already solved" lacks empirical rigor. Generalizing from a saturated ~1% gap on a simple dataset to completely question the necessity of a dual-branch architecture for complex, real-world domain shifts commits an overgeneralization.
> > > - Claimed INR T20: Also see Table1 items v&vi; overlooks explicit statement that Table1 were selected from extensive pilot studies (Section3.3).
> > > # To 4
> > > - Stated in 1-⑥⑦.
> > > - CSE is also a theoretically necessary complement to UGC. It mathematically enforces class-specific decoupling, which provides structural guarantees for mitigating feature confusion that absolute accuracy metrics at saturation simply cannot fully capture.
> > >
> > > We sincerely thank the reviewer, and would be deeply grateful if you might reconsider your overall evaluation in light of these new insights. We believe VILA's rigorous dual-branch decoupling provides a highly effective and theoretically grounded paradigm for PTM-based CIL.

---

### Official Review · Reviewer_LBuE · 2026-03-13

**Soundness:** 3
**Presentation:** 2
**Significance:** 3
**Originality:** 2
**Overall Recommendation:** 4
**Confidence:** 4

**Summary:**

This paper provides a new pre-trained based CL method VILA and studies the failure modes of analytic based solutions, which is identified as repreentation rigidity. VILA is a dual-branch framework that fuses a task-adapted ViT branch with a frozen CLIP via L2-normalized concatenation, and implements a decision-level candidate semantic enhancement using CLIP prototypes on a top-k set. Evaluated over eight benchmarks, and multiple sequence lengths, the effectiveness of VILA is well supported.

**Compliance With Llm Reviewing Policy:**

Affirmed.

**Final Justification:**

My main concerns have been well-addressed. Due to the current limited empirical scope of the work, I will maintain the original score as weak accept.

**Key Questions For Authors:**

please see the weakness part.

**Limitations:**

yes.

**Strengths And Weaknesses:**

**Strength**:

This paper presents a well-structured and highly coherent narrative that formally identifies the "representation rigidity" bottleneck in analytic class-incremental learning. To address this, the authors propose an innovative dual-branch framework coupled with an efficient semantic refinement mechanism, balancing recursive stability with cross-modal plasticity. The methodology is rigorously validated across diverse and challenging scenarios, including fine-grained and long-sequence tasks, demonstrating practical improvement in the trade-off between accuracy and efficiency.

------


**Weakness**:

1. It seems that the authors overclaim the necessity and function of L2 nomalization pluse concatenation as "unified geometric calibration", and this step is generally standard and lack significant novelty.

2. Can the authors provide more comprehensive ablation study across other benchmarks to further validat the contribution of each components?

3. (minor issue) an overall method diagram can greatly enhance the readability and clarity of the manuscript.

---

> ### Author Rebuttal · Authors · 2026-03-30
>
> We sincerely thank the reviewer for the constructive feedback and the positive overall evaluation. We have carefully considered all your comments, and have addressed them point-by-point below. We hope them adequately resolve your concerns.
> ## Weakness1 UGC novelty
> We agree that L2-norm+concat is a common and standard operation, and we certainly do not claim them as a new invention. However, we want to clarify that **our core contribution here is about diagnosing and fixing a hidden geometric conflict that affects the Analytic CIL system**.
>
> ① When only a single branch is used (ViT/CLIP) in Analytic CIL, applying L2-norm has a small impact on the final accuracy. But when we fuse heterogeneous features, L2-norm is necessary to keep the dual-branch analytic system from collapsing. ViT-Adapter features live in an unbounded space (huge numerical values), while frozen CLIP features live in a hypersphere (smaller values).
>
> ② We believe **using a simple mathematical operation to solve a deep structural problem is an elegant approach, much like the residual connection in ResNet** for example. The residual connection is fundamentally just a basic addition sign ("+"), but it brilliantly solves the issue of vanishing gradients. We looked at our problem the exact same way.
>
> ③ We wish to highlight that algorithmic originality does not strictly require inventing mathematically complex new formulas. Often, **substantial novelty can also stem from providing "novel insights by evaluating existing methods" (Section3) and using "creative combinations of existing ideas" (Section4) to resolve them**. Our work discover that ViT-adapter features mathematically misalign with the universal CLIP features within analytic solvers, the targeted application of UGC represents exactly this kind of problem-driven insight.
>
> We also wish to emphasize that Dual-Branch (DB) design (Section3&Section4.1) to solve representation rigidity is our primary and the most important contribution.
>
> The necessity of UGC is **fundamentally inseparable from this DB design** (Feature-level calibration = DB+UGC). Without heterogeneous features, there would be no geometric misalignment to calibrate.
>
> Please also see response to Reviewer QYHV-Weakness1&Question1 for further novelty clarification.
>
> Please also see response to Reviewer czxT-W2&Q1, QYHV-Weakness3&Question3, which clarifies the importance of DB+UGC.
> ## Weakness2 All benchmark ablation
> |$A_T/\bar{A}$|CIFAR T10|ImageNet-R T20|CUB T10|UCF T10|Food T10|SUN T30|
> |-|-|-|-|-|-|-|
> ||90.79/94.22|76.72/82.15|88.72/93.10|97.50/98.73|86.74/91.36|78.55/85.52|
> |CSE| 90.95/94.33|78.43/83.73|89.02/93.21|97.54/98.77|87.16/91.65|79.13/85.92|
> |DB| 91.74/94.82|82.57/87.61|89.78/93.23|98.37/99.36|88.13/92.32|81.35/87.72|
> |DB+UGC| 91.87/95.01|84.95/89.50| 89.86/93.56|98.90/99.43|89.01/92.97|83.60/89.32|
> |DB+UGC+CSE|91.94/95.09|85.28/89.81|90.08/93.64|98.94/99.45|89.30/93.04|83.73/89.43|
>
> Combined the table above with Table 4 (Aircraft and Cars) in our original manuscript, we now provide ablation results across all 8 diverse benchmarks. We will include them in the main text or Appendix considering page limitation.
> ## Weakness3 Provide overall method diagram
> We have designed a overall method diagram, which is provided in **[[this link]](https://anonymous.4open.science/r/Figures-84FE/)**. The caption is: From observation to solution. Left: feature space trilemma. The specialized ViT-Adapter branch collapses into a rigid subspace, which creates a geometric misalignment with the universal CLIP hypersphere and fails to cover future-task features. Right: VILA asymmetric dual-branch architecture. It integrates a frozen universal branch to mitigate stability-plasticity dilemma (Obs1 & Obs2). The UGC module projects heterogeneous features onto a unified manifold to address misalignment (Obs3). The CSE module leverages text priors to rectify decision boundaries. Solely training on Task 1 and updating classifiers via analytic learning ensures extreme efficiency.
>
> We have included this figure in the main text of the revised manuscript.

---

> > ### Author Rebuttal · Reviewer_LBuE · 2026-04-04
> >
> > Thanks for the rebuttal efforts and all my main concerns have been well addressed. Please incorporate the additional results, figures and clarifications in the revised manuscript. I will maintain my positive score.

---

> > > ### Author Response · Authors · 2026-04-06
> > >
> > > We are very pleased that our rebuttal and the newly provided evidence have fully addressed all of your concerns.
> > >
> > > We deeply appreciate your constructive suggestions, which have undoubtedly enhanced the clarity and completeness of our paper. We ensure that the comprehensive ablation results across all 8 benchmarks, the overall method diagram and novelty clarifications, have been incorporated into the revised manuscript.
> > >
> > > Thank you once again for your time, your valuable guidance, and your positive support for our work.

---

### Decision · Program_Chairs · 2026-04-30

**Decision:**

Accept (regular)

**Comment:**

## 1. Summary
This paper studies analytic class-incremental learning with pre-trained models and identifies **representation rigidity** as a key bottleneck. To address this, it proposes **VILA**, a dual-branch framework that combines a task-adapted visual branch with a frozen semantic branch, together with feature-level geometric calibration and decision-level semantic refinement. The paper argues that this design improves plasticity while preserving the efficiency advantages of analytic learning.

## 2. Reviewer evaluation and concerns
The reviews were overall positive. Reviewers appreciated the clear diagnose-then-design structure, the strong empirical coverage across multiple benchmarks and sequence lengths, and the practical relevance of improving analytic CIL under strict efficiency constraints. Several reviewers found the motivation meaningful and the experimental performance convincing, especially in fine-grained and long-sequence settings.

The main concerns were:
- the **algorithmic novelty** of the calibration components is limited when viewed in isolation, since they rely on standard normalization, concatenation, and reranking-style operations;
- the **fairness of comparisons** needed to be clarified because VILA uses a dual-branch architecture with additional pretrained components;
- the paper initially needed **stronger ablations and controls**, especially to isolate the contribution of the dual-branch design, UGC, and CSE;
- some reviewers also raised questions about parameter/FLOP cost, possible data-leakage concerns from CLIP, and the need for broader discussion of related work.

## 3. Discussion
The rebuttal substantially strengthened the paper. The authors added broader ablations across all benchmarks, provided the requested control with CLIP-augmented baselines, included FLOP/parameter analysis, and presented additional experiments replacing CLIP with DINOv2 to address the concern that gains might come mainly from CLIP’s pretraining corpus. These additions clarified that the main contribution is not a new primitive operation, but a **problem-driven dual-branch analytic CIL design** with principled feature calibration and semantic correction.

While some individual components are simple and that the marginal gain of the decision-level module is limited, the overall framework is technically coherent, empirically strong, and now much better supported after rebuttal. The paper makes a useful contribution to analytic CIL by showing how to combine efficient recursive learning with semantic priors in a way that consistently improves performance across diverse settings.